# RL Grokking Recipe: How Does RL Unlock and Transfer New Algorithms in LLMs?

**Yiyou Sun**[1], **Yuhan Cao**,[*] **Pohao Huang**[1], **Haoyue Bai**[2], **Hannaneh Hajishirzi**[3,4],
**Nouha Dziri**[4][†] **Dawn Song**[1][†]
[1]University of California, Berkeley, [2]University of Wisconsin, Madison,
[3]University of Washington, [4]Ai2

## Abstract

It remains an open question whether LLMs can acquire or generalize *genuinely new reasoning strategies*, beyond the sharpened skills encoded in their parameters during pre-training or post-training. To attempt to answer this debate, we introduce *DELTA* — Distributional Evaluation of Learnability and Transferrability in Algorithmic Coding, a controlled benchmark of synthetic coding problem families designed to probe two fundamental aspects: *learnability*—can LLMs, through reinforcement learning (RL), solve problem families where pretrained models exhibit failure with large enough attempts (pass@K=0)?—and *transferability*— if learnability happens, can such skills transfer systematically to out-of-distribution (OOD) test sets? Unlike prior public coding datasets, DELTA isolates reasoning skills through templated problem generators and introduces fully OOD problem families that demand novel strategies rather than tool invocation or memorized patterns. Our experiments reveal a striking **grokking** phase transition: after an extended period with near-zero reward, RL-trained models abruptly climb to near-perfect accuracy. To enable learnability on previously unsolvable problem families, we explore key training ingredients such as staged warm-up with dense rewards, experience replay, curriculum training, and verification-in-the-loop. Beyond learnability, we use DELTA to evaluate transferability or generalization along *exploratory*, *compositional*, and *transformative* axes, as well as cross-family transfer. Results show solid gains within families and for recomposed skills, but persistent weaknesses in transformative cases. DELTA thus offers a clean testbed for probing the limits of RL-driven reasoning and for understanding how models can move beyond existing priors to acquire new algorithmic skills. Code is available in `https://github.com/sunblaze-ucb/rl-grok-recipe`.

## 1 Introduction

A central question for RL on language models is whether it merely sharpens latent skills or enables genuinely new reasoning. Some argue RL only refines existing heuristics embedded in the model's parameters (Yue et al., 2025; Wu et al., 2025), while others see it as a way to unlock emergent problem-solving (Liu et al., 2025b;a). We make this debate testable using two criteria: *learnability*, which asks if RL can instill a procedure the model could not previously execute; and *generalization*, which asks if that procedure transfers to diverse Out-of-distribution (OOD) cases rather than memorized patterns. Addressing these questions requires a dataset with tightly controlled train–test splits that can systematically probe both properties.

**Why controlled problem families matter?** Uncontrolled open benchmarks in math/coding (e.g., *Numina-Math* (Li et al., 2024), *DeepMath* (He et al., 2025), *OpenCodeReasoning* (NVIDIA, 2025)) mix topics and difficulty, blurring the line between capability sharpening and genuine acquisition. Controlled synthetic families remove these confounds: we can precisely vary distributions and difficulty, attribute gains to specific skills, detect phase transitions, and systematically test transfer to OOD variants.

---

[*]Main dataset contributor, independent researcher.
[†]Indicates equal advising role in alphabetical order.

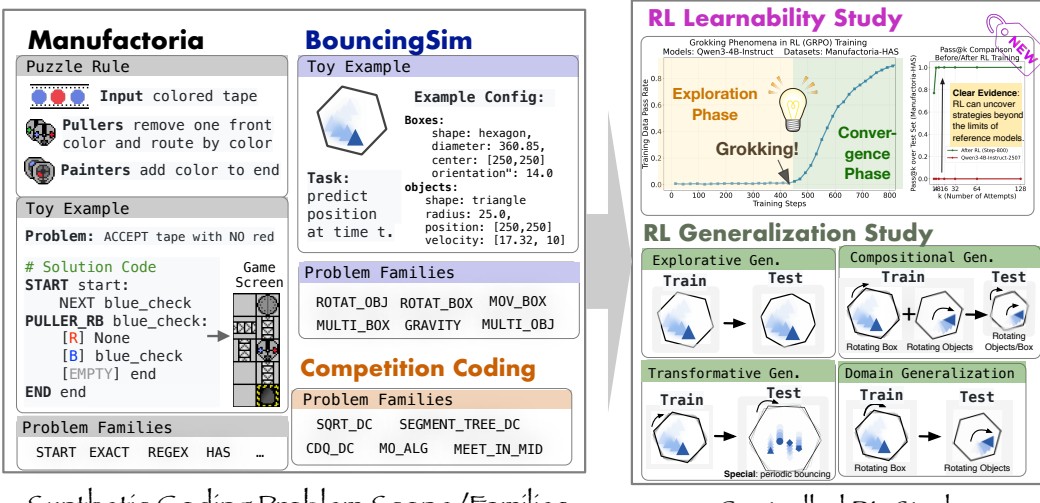

Figure 1: Overview of DELTA with controlled RL studies. *Left*: Synthetic Programming Problem families—Manufactoria with custom syntax and puzzle-like rules, BounceSim with physical simulation, etc. *Right*: Controlled RL experiments. *Top*: Learnability shows grokking, where RL shifts from long exploration to sudden convergence, uncovering strategies beyond reference models. *Bottom*: Generalization extends OMEGA (Sun et al., 2025) across four axes—Exploratory, Compositional, Transformative, and Domain-level—testing adaptation to harder or recombined tasks.

**Why programming problems?** GRPO/PPO pipelines typically rely on a pass/fail reward: a perfect solution earns +1, anything else earns 0 Guo et al. (2025). This sparsity can stall learning on hard families. In math, grading intermediate steps is expensive and hard to scale. Programming, however, naturally supplies fine-grained feedback through test cases, which act as dense rewards. A practical approach is to start training with test-case–based rewards to encourage partial progress, then transition to a binary outcome reward to lock in exact solutions. This staged scheme is crucial for helping LLMs acquire genuinely new procedural strategies, and while coding offers a uniquely scalable setting, the underlying insight of using intermediate signals before enforcing strict correctness may apply to other reasoning-heavy domains such as math or formal logic.

To address this need, we introduce **DELTA**, a controlled yet diverse benchmark for programming problems. DELTA consists of synthetic problem families drawn from different domains, each generated from templated problem generators, allowing us to study phenomena such as difficulty scaling, knowledge transfer, and learnability in a clean and isolated setting.

**RL Learnability Study.** We reveal an underexplored **grokking** phenomenon during RL training. While recent works argue that RL cannot exceed the limits of its reference model (Yue et al., 2025; Wu et al., 2025), our evidence suggests otherwise. On hard problems where the base model achieves pass@K = 0[1], standard RL with binary rewards collapses due to the absence of positive signals. By contrast, a staged regime—warming up with fine-grained proxy rewards before switching to strict pass/fail—first guides exploration into a region where full solutions become reachable, then sharpens these into verified completions, producing a long exploratory plateau followed by sudden grokking to near-perfect accuracy (Figure 1, top-right).

**RL Generalization Study**. DELTA extends OMEGA's controlled tests along three axes aligned with Boden's creativity typology (Boden, 1998): (1) *Exploratory*—extend known skills within a family (e.g., hexagon to octagon); (2) *Compositional*—combine previously separate skills (e.g., bouncing ball with both rotating obstacles and boxes); (3) *Transformative*—discover unconventional solutions (e.g., special initial states that guarantee periodicity). Our results show that RL-trained models generalize to harder and composed variants, but performance drops with complexity, and transformative cases remain the most challenging.

---

[1]Here pass@K refers to a large value of K (e.g., 128). Thus, pass@K = 0 indicates that the model fails to solve the task even after many sampled attempts.

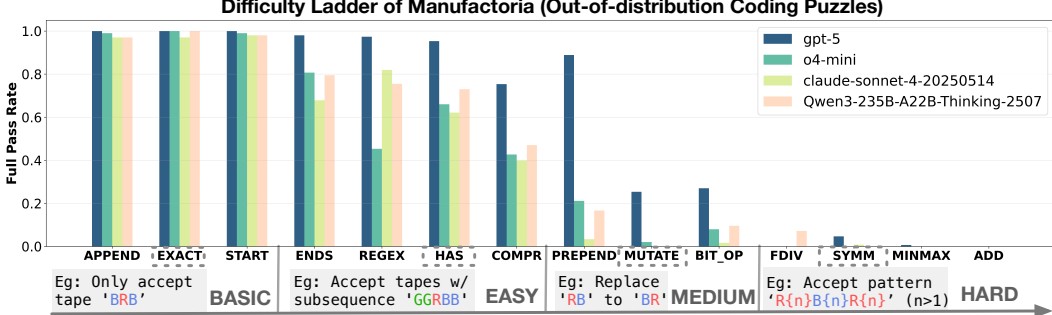

Figure 2: The Manufactoria difficulty ladder. 14 problem families are grouped into Basic, Easy, Medium, and Hard levels according to average performance across four popular LLMs. Each test split contains 20–50 problems, and *full pass rate* are averaged over 4 independent runs.

**Main contributions.** 1) **A controlled dataset (DELTA)**: We design a suite of synthetic programming problem families that isolate reasoning skills, enabling clean tests of learnability (can RL unlock procedures absent in the base model) and generalization (do these procedures transfer systematically to OOD cases). Unlike prior coding or math datasets, DELTA introduces fully OOD problems (Manufactoria) and richly graded rewards, avoiding tool-based shortcuts and data confounds.

2) **Sharpening or discovery, depending on setup**: We provide clear evidence that RL is not limited to sharpening existing abilities in reference models. On hard families where base models fail (pass@K=0), staged training with dense-to-binary rewards produces a grokking phase transition — a sudden leap from failure to mastery — showing that RL can indeed discover strategies the base model could not execute. At the same time, in easier regimes or with weaker setups, RL primarily sharpens existing skills. Which outcome emerges depends critically on the reward design, data mix, task hardness, and training recipes.

3) **Three-axis generalization analysis**: We evaluate how these learned strategies transfer along exploratory, compositional, and transformative axes. Results show strong generalization in exploratory and recomposed cases, but persistent failures in transformative shifts, highlighting both the promise and limits of RL-driven reasoning and the generalization challenges we must work on.

## 2 DELTA: CONTROLLED PROGRAMMING PROBLEM FAMILIES

We operationalize *learnability* and *generalization* with **DELTA**, a controlled suite of synthetic programming families.

**From OMEGA to DELTA.** OMEGA (Sun et al., 2025) offers 40 synthesizable math families to study exploratory, compositional, and transformative generalization in the spirit of Boden (Boden, 1998). DELTA complements this by shifting to programming, where templated generators yield automatically verifiable tasks with tunable difficulty and clean distributional controls. Compared to OMEGA, DELTA further provides unique benefits and improvements: a) **Novel OOD problem family.** Math tasks in OMEGA remain within familiar domains (e.g., algebra, geometry), which can plausibly appear in pretraining corpora. In contrast, DELTA includes a hand-crafted out-of-distribution (OOD) problem scope called `Manufactoria`, which uses entirely novel program syntax and problem-solving strategies. b) **Harder to shortcut with tools.** Many synthetic math items can be solved by executing Python (e.g., computing a matrix rank). In DELTA, the target is the program itself: models must synthesize a correct solution rather than delegate computation to external tools. c) **Rich reward signal.** Programming enables cheap, graded feedback via per–test case pass rates, which supports staged training (dense reward then binary full-pass reward).

In DELTA, we design problems from five major scopes, illustrated in Figure 1. We next introduce the problem families in detail.

### 2.1 MANUFACTORIA (OUT-OF-DISTRIBUTION PROBLEMS FOR LEARNABILITY STUDY)

*Manufactoria* is a classic Flash game (2010) in which players build automated factories to sort robots based on their colored tape patterns. The underlying logic resembles constructing finite-state automata or tag systems using two special node types (`puller`, `painter`). While the original game

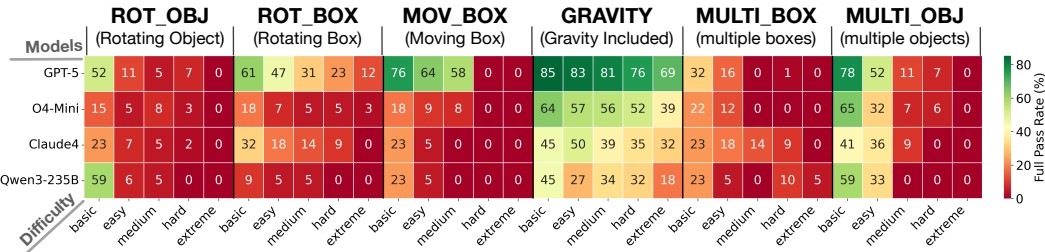

Figure 3: Full-pass rate (%) on *BouncingSim* by model, family (ROT_OBJ, ROT_BOX, MOV_BOX, GRAVITY, MULTI_BOX, MULTI_OBJ), and difficulty tier (BASIC→EXTREME). Warmer colors denote higher accuracy; cell values are mean full-pass rates per split over 4 runs on 50 test problems each.

is implemented in 2D space, we re-formalize it into a custom programmatic syntax, as illustrated in Figure 1. Details are provided in Appendix A.1.

**Justified OOD-ness.** This task is OOD for several reasons: a) The original game solutions were stored only as images on legacy websites. Our converted program syntax is entirely novel and unavailable to any LLM during pretraining; b) We do not reuse existing game challenges. Instead, we design new problem families inspired by the mechanics but synthesized by the authors, and these are entirely unseen to LLMs; c) The puzzle strategies are qualitatively different from conventional programming or Turing-machine tasks. With only two available node types with limited functionality, solving requires distinctive reasoning patterns not captured by standard coding strategies.

**A scalable difficulty ladder.** In total, we construct 14 synthetic problem families. For example, the family tagged HAS (Figure 2) requires accepting tapes that contain a subsequence such as GGRBB, which can be synthesized by using arbitrary color strings. *Manufactoria* is organized into BASIC → EASY → MEDIUM → HARD tiers, enabling matched studies across model scales. BASIC/EASY families (e.g., START, EXACT) suit small models (e.g., 1.5B, 4B) for learnability, while MEDIUM/HARD families require more advanced insight and are appropriate for probing SOTA systems (e.g., GPT-5–class). Because the syntax and families are novel, *Manufactoria* also serves as an OOD benchmark for open LLMs, enabling apples-to-apples comparisons with SOTA LLMs on truly novel tasks. Medium tasks expose a larger gap: only GPT-5 achieves non-trivial success, while other models collapse near zero. Hard families remain unsolved across the board, underscoring the sharp transition in difficulty and the limits of the current model.

## 2.2 BOUNCINGSIM (2D SIMULATION PROGRAMMING TASKS FOR GENERALIZATION STUDY)

We include a widely used community test—a 2D bouncing-ball simulation program—often treated as a proxy for geometry-aware reasoning in LLMs (Wiggers, 2025). The goal is to synthesize a program that simulates elastic collisions in polygonal containers and returns the exact object state at a queried timestamp; strong solutions require precise collision detection/response and numerically stable integration.

**Task design.** To replace informal, visually judged demos with a rigorous benchmark, we make the task: (a) *verifiable*—each prompt specifies a deterministic initial state (positions, velocities, container geometry); the program must output the object's location at a target time and is scored against an oracle; (b) *synthesizable*—instances are generated by varying the configuration in Figure 1, with ground-truth trajectories produced by *Box2D*[2]; (c) *composable*—single-skill families (e.g., ROT_BOX, ROT_OBJ) can be combined into multi-skill families (e.g., ROT_BOX_OBJ); and (d) *difficulty-controlled*—we vary polygon vertex counts, object speeds, box motion, gravity, and the number of objects/boxes to create BASIC→EASY→MEDIUM→HARD→EXTREME tiers. Detailed configurations are provided in Appendix A.

**Generalization axes.** To align explicitly with the three generalization axes defined in OMEGA (Sun et al., 2025), as exemplified in Figure 1: a) *Exploratory generalization*: Training problems feature standard box sizes with relatively sparse collisions, while test problems use smaller containers that induce denser and more frequent collisions. b) *Compositional generalization*: Training isolates distinct skills—handling rotating boxes (ROTAT_BOX) and rotating objects (ROTAT_OBJ). Testing

---

[2]https://box2d.org/

then evaluates the combined scenario (`ROTAT_BOX_OBJ`), where both the box and the object rotate simultaneously, requiring the model to integrate the two skills. c) *Transformative generalization*: Training covers common variants such as `ROTAT_BOX`, but testing introduces qualitatively different dynamics—for example, special initial conditions that yield perfectly periodic bouncing trajectories (e.g., an object oscillating vertically with no horizontal drift). Further examples and details of these generalization setups are provided in Appendix A. **Evaluation results.** Figure 3 summarizes full-pass rates across six families—`ROT_OBJ`, `ROT_BOX`, `MOV_BOX`, `GRAVITY`, `MULTI_BOX`, `MULTI_OBJ`—and five difficulty tiers for four representative models. GPT-5 leads overall, but accuracy degrades with difficulty and composition: `MULTI_BOX` is challenging even at BASIC (∼30%), and `MULTI_OBJ` drops sharply—from ∼80% at BASIC to ∼10% by MEDIUM. Other LLMs trail substantially—typically ≤30–40% on the easy-to-medium tiers and near-zero on HARD/EXTREME and most compositional settings. Overall, *BouncingSim* represent a valuable testbed for understanding what these models can and cannot do; whether they reinforce existing skills or discover new ones; by enabling systematic study of learnability and generalization.

## 2.3 COMPETITION CODING PROBLEM FAMILIES

We add three real-world domains—competitive programming, SQL, and LEAN. Although not strictly OOD (given their online popularity), they remain challenging (e.g., *gpt-5-high* reaches only 2% on hard-tier LiveCodeBench-Pro (Zheng et al., 2025)). We include them in DELTA to expand seed problems into fully controlled families that support learnability and generalization studies. A brief construction overview appears in the main text, with details in Appendix A. Each family groups problems sharing the same core algorithm (e.g., *Mo's algorithm*, *CDQ divide-and-conquer*), and is named after that algorithm. For each family, we: (1) gather 5–7 seed tasks verified to use the target algorithm; (2) perturb their contexts by relying on an expert-provided solution strategy and background, then use LLM to change narrative surface while preserving the solution; and (3) filter and verify by requiring a brute-force solution to pass all tests, ensuring perturbation consistency. We release 5 families (∼500 items each).

## 3 LEARNABILITY STUDY: CAN RL UNCOVER NEW STRATEGIES AND HOW TO ACCELERATE IT?

A central debate in recent research concerns whether reinforcement learning (RL) can endow models with reasoning abilities beyond those of their base model.

**The skeptical view.** Yue et al. (2025) argue that although RLVR-trained models outperform their base models at small $k$ (e.g., $k = 1$), the base models achieve equal or superior pass@k performance when $k$ is large. Their coverage and perplexity analyses suggest that reasoning capabilities are ultimately bounded by the base model's support. Similarly, Wu et al. (2025) provide a theoretical argument that RLVR cannot extend beyond the base model's representational limits.

**The optimistic view.** In contrast, Liu et al. (2025b) demonstrates that ProRL can expand reasoning boundaries on tasks where the base model performs poorly—specifically in letter-formed 2D puzzles from Reasoning Gym (Stojanovski et al., 2025).

**Our contribution: a clean testbed and clear evidence for RL enable grokking in LLMs.** Existing evidence in favor of RL's generalization often comes from large, heterogeneous training corpora. This makes it difficult to isolate why and how RL might discover novel strategies. To address this, DELTA offers a controlled environment: synthetic problem families that are both out-of-distribution (requiring novel strategies) and internally consistent (free of data confounds). We focus on the `Manufactoria-HAS` family (742 training / 100 test instances), where the reference model *Qwen3-4B-Instruct-2507* achieves **0% full pass rate at pass@128**. As shown in Figure 4, our staged RL training strategies enables the model to fully solve this family, achieving 100% full pass rate. Next, we detail how this is made possible.

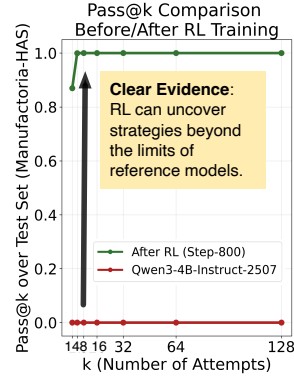

Figure 4: Pass@k comparison before and after RL training on the `Manufactoria-HAS`.

## 3.1 BASIC SETUP

Unless otherwise specified, the reference model is *Qwen3-4B-Instruct*. Training and testing datasets are drawn from single or combined problem families introduced in Section 2. By default, each training step consists of 48 prompts with 16 rollouts. The learning rate is set to $5 \times 10^{-7}$. For code training, the default reward signal is *full pass*, a binary indicator of whether a program passes all test cases. In later experiments, we also consider *per-test pass rate* as the reward signal, measuring the fraction of test cases passed. A more detailed experiment setup parameter descriptions are included in Appendix B. We also provide complementary experiments with alternative model families, sizes, and problem domains in Appendix C.1.

## 3.2 HOW TO SOLVE "PASS@K=0" TASKS WITH RL?

The skeptical position that RL cannot exceed the boundaries of the base model is understandable for a simple reason: GRPO (Guo et al., 2025) depends on reward differences across rollouts. If no rollout ever succeeds (as in "pass@K=0" tasks), there is no gradient signal to learn from. Indeed, as Figure 5(a) shows, naïve GRPO training stagnates. Thus, the central challenge is:

*If no rollout achieves a full pass, how can RL propagate a meaningful learning signal?*

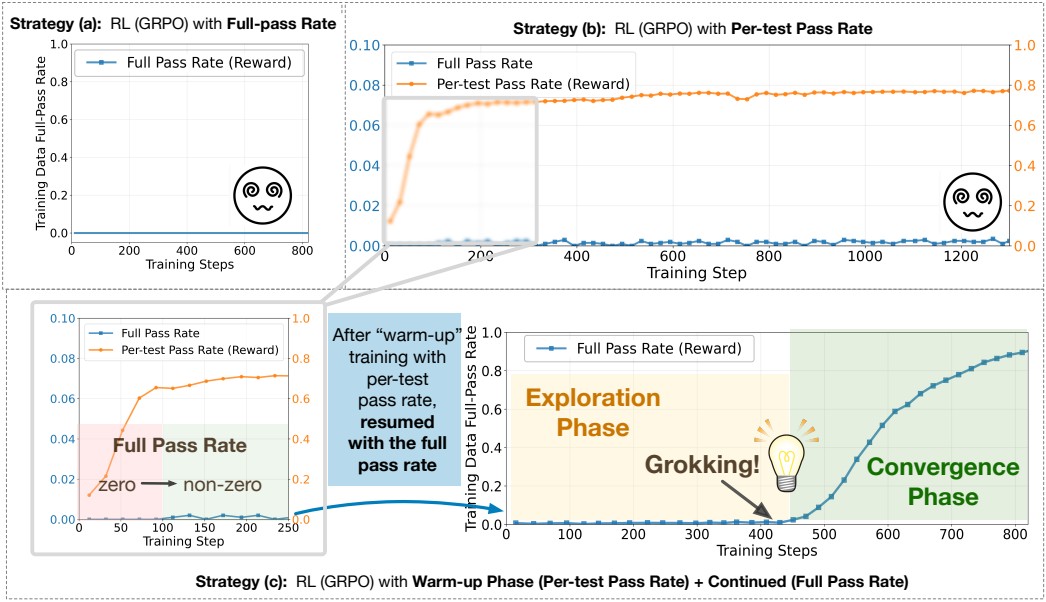

Figure 5: Comparison of strategies solving "pass@K=0" tasks. (a) Directly optimizing for full-pass rate under GRPO fails. (b) Training with a per-test pass rate provides a smoother reward but quickly saturates. (c) A two-phase training—warming up with per-test pass rate, then switching to full-pass reward. All training is performed on `Manufactoria-HAS` family and the reference model *Qwen3-4B-Instruct-2507*.

**Per-test pass rate training.** One solution is to exploit partial credit. Instead of the all-or-nothing full pass rate (reward = 1 only if all test cases pass), we use a finer-grained per-test pass rate, a continuous reward in $[0, 1]$. As Figure 5(b) shows, this signal provides initial learning traction. However, it quickly saturates after ~100 steps, and the full-pass rate remains negligible (<0.01%).

**Warm-up phase.** Even though it can not serve as a full surrogate loss, we find that the per-test pass rate can serve as an important warm-up stage that pushes the model out of the all-zero region. As shown in Figure 5(a), this signal allows the model to move beyond the all-zero region: although the full-pass rate remains $< 1\%$, the model begins to accumulate positive gradients.

**Exploration and grokking.** From this warm-up checkpoint, we switch to RL with the binary full-pass reward. Figure 5(b) illustrates the dynamics: For ~450 steps, the model remains in an *exploration phase*, with full-pass rate still $< 1\%$. After a sudden **grokking moment**, the model discovers the key strategy to solve the family. Training then enters a *convergence phase*, where RL sharpens and consistently reinforces the successful reasoning path. At convergence, the RL-trained model achieves

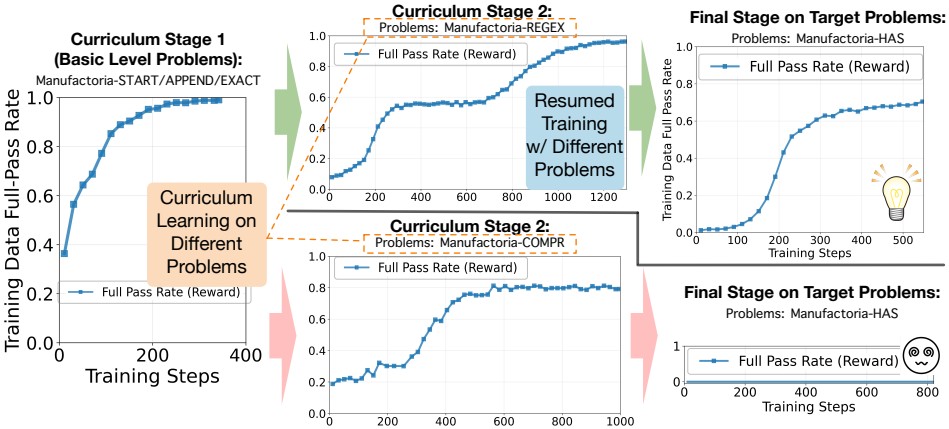

Figure 7: Contrast of the two-stage curriculum learning for `Manufactoria-HAS`. Models first train on basic problems (`START/APPEND/EXACT`) before branching into one of two intermediate curricula: (i) Stage 2–`REGEX`, which leads to successful transfer and high pass rates on the target `HAS` family, or (ii) Stage 2–`COMPR`, which fails to transfer and plateaus at low performance.

nearly a 100% absolute improvement in pass@k compared to the reference model (Figure 4). We also observe this phenomena with other model families, sizes, and problem domains in Appendix C.1

## 3.3 ATTEMPTS TO ACCELERATE RL GROKKING

A natural follow-up question is how to shorten the exploration phase and enable grokking to emerge earlier. We examine the following strategies:

**Experience replay**. The long exploration phases mainly stem from the sparsity of positive reward signals. A natural way to alleviate this is to retain successful reasoning traces and reinsert them into future rollouts—a technique known as experience replay (Zhang et al., 2025), closely related to expert iteration (Anthony et al., 2017). In our experiments, we log successful traces in each sampling round and, when the same query reappears, append up to three of the most recent successful traces to the rollout. As shown in Figure 6, experience replay does help the model grok at an earlier stage. However, its convergence speed is still slower than the baseline GRPO algorithm, likely because the reused traces are off-policy.

**Feedback-in-the-loop**. Another plausible strategy is to directly include failure feedback in the generation process, encouraging the model to improve its full pass rate earlier. We achieve this by replacing the `EOS` token with feedback (e.g., failure test cases) and letting the model continue generating. As shown in Figure 6, applying this feedback-in-the-loop once can indeed expedite the grokking moment. However, it also reduces training stability, likely due to the off-policy injection of feedback tokens. A common failure case is that the model, even after receiving explicit feedback, persists in its original (incorrect) solution.

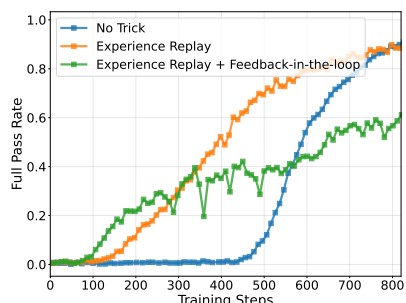

Figure 6: Comparison of training strategies for accelerating RL grokking. "No Trick" denotes the standard training setup as in Figure 5(b), "Experience Replay" logs and reuses successful traces, and "Experience Replay + Feedback-in-the-loop" further injects verifier's feedback into the inference.

## 3.4 MORE INVESTIGATION INTO THE WARM-UP PHASE

**Selective curriculum learning as an alternative.** A natural question is whether the warm-up effect can be achieved through curriculum learning across problem families. To explore this (Figure 7), we designed a three-stage curriculum training. After training on basic families (`START/APPEND/EXACT`), models were exposed either to Stage 2–`REGEX` or Stage 2–`COMPR` before transferring to the target `HAS` tasks. These two problem families have similar difficulty levels

according to Figure 2. Despite similar difficulty, the outcomes diverge: the `REGEX` curriculum leads to successful transfer and near-complete mastery of `HAS` at final RL stage, while the `COMPR` curriculum fails to progress beyond low pass rates. This difference can be traced to task compatibility—both `REGEX` and `HAS` revolve around detecting or matching subpatterns (e.g., "accept tapes with pattern `(BRB)`$^+$`(RR)`$^*$" vs. "accept tapes with subsequence `GGRBB`"), whereas `COMPR` emphasizes numerical interpretation and branching tests (e.g., "treat color `B` as 1 and `R` as 0, accept if the number $\geq 27$"). These results suggest that effective curricula must not only control difficulty but also align structurally with the target family. While curriculum learning can thus be highly effective, its success depends on finding suitably related families to bridge the reasoning gap—something that is not always feasible. In contrast, warm-up training with dense rewards remains broadly useful as it does not require additional family design or mixing.

**Warm-up Helps Beyond the "pass@k=0" Regime** Even when the base model exhibits a small but non-zero success rate ($pass@k = \epsilon > 0$), a brief per-test–reward warm-up improves stability and speed. Empirically, we observe faster and smoother convergence compared to training full-pass from scratch (see Appendix C.2).

**Limitation.** It is important to note that not every problem family can be "unlocked" by warm-up training. For instance, as shown in Figure 8, even when using per-test pass rate rewards, the model fails to escape the all-zero regime on the harder `Manufactoria-PREPEND` family. The per-test signal rises modestly but quickly saturates, while the full-pass rate remains stuck at zero throughout training. This suggests that warm-up with per-test pass rate training is not a universal recipe: its effectiveness depends on the model's capacity and difficulty of the target family.

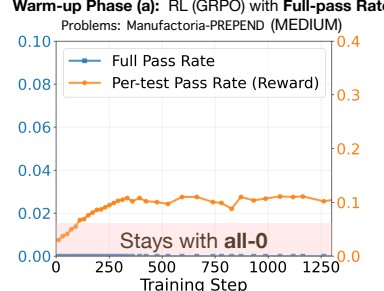

Figure 8: Warm-up training on the harder `Manufactoria-PREPEND` family.

## 4   GENERALIZATION STUDY

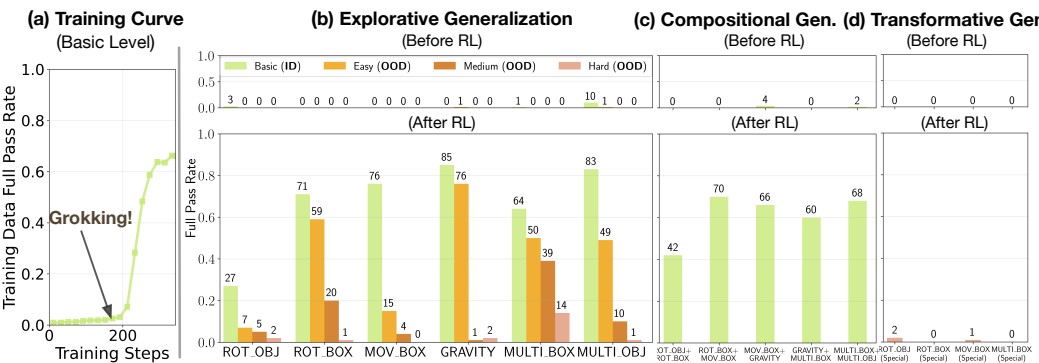

Figure 9: **Generalization Study on BOUNCINGSIM**. (a) Training full-pass rate on the Basic-level mixture (6 families, 1k each) for *Qwen3-4B-Instruct* with binary full-pass reward shows a sharp *grokking* jump near step 200. (b) *Explorative generalization:* Before RL (top) the model rarely solves any OOD cases; after RL (bottom) it transfers to Easy/Medium/Hard variants with diminishing gains as difficulty increases (bars aggregate 6 families × 4 tiers; 100 prompts per cell, averaged over 4 runs). (c) *Compositional generalization:* Zero-shot composition of skills. (d) *Transformative generalization:* Qualitatively new dynamics (e.g., special periodic trajectories) remain near zero after RL. Results are averaged over 4 runs.

**Setup.** We study how far the learned programmatic skills transfer beyond the training distribution. Unless noted, the reference model is *Qwen3-4B-Instruct*. We train on a Basic-level mixture of six single-skill families—`ROT_OBJ`, `ROT_BOX`, `MOV_BOX`, `GRAVITY`, `MULTI_BOX`, `MULTI_OBJ`—with 1k instances per family (6k total). Because the base model has non-zero full-pass on some basic instances, we directly optimize a *binary full-pass reward* (all tests pass) for 300 gradient steps; all other hyperparameters follow Section 3. Evaluation spans three axes—*explorative*, *compositional*, and *transformative*—and reports *full pass rate* (fraction of prompts for which the synthesized program exactly matches the oracle on all unit tests). For explorative generalization we

consider four difficulty tiers (Basic=ID, Easy/Medium/Hard=OOD) crossed with the six families; each bar in Figure 9 aggregates. More detailed setup is in Appendix B.

**Training dynamics (Fig. 9a).** We again observe a sharp *grokking* phase transition: after a long plateau of near-zero reward, performance on the training mixture jumps around the step 200 to 0.7 full pass rate, indicating the emergence of stable simulation code that handles elastic collisions.

**Generalization results (Fig. 9b–d).** RL-trained models transfer beyond the training distribution, but with varying success across axes. In *explorative generalization*, performance is strong on Basic (ID, 70–85%) and carries over to Easy (50–75%), though gains shrink on Medium (15–50%) and nearly vanish on Hard (single digits). For *compositional generalization*, the model demonstrates surprising skill integration: unseen combinations such as `ROT_BOX+MOV_BOX`, `MOV_BOX+GRAVITY`, and `MULTI_BOX+MULTI_OBJ` achieve 60–70% full-pass (vs. near-zero before RL), in contrast to the weak compositional transfer reported in OMEGA (Sun et al., 2025). We attribute this to coding tasks composing *structurally* (merging simulation modules) rather than *strategically* (inventing new reasoning steps). Finally, in *transformative generalization*, models remain near zero on qualitatively novel dynamics such as perfectly periodic or degenerate trajectories, which demand the discovery of new invariants and align with the persistent difficulty of transformative math generalization.

**Takeaways.** RL discovers executable simulators that (i) transfer well to parametric shifts and (ii) compose across skills, but (iii) struggle when the test distribution demands qualitatively different solution schemas. Coding tasks appear more amenable to structural composition than symbolic math, yet transformative "schema creation" remains an open challenge. Figure 9 summarizes these trends.

## 5 RELATED WORK

**Coding benchmarks and synthetic datasets.** Human written or collected coding benchmarks like APPS (Hendrycks et al., 2021), CodeContests (Li et al., 2022), HumanEval (Chen et al., 2021), MBPP (Austin et al., 2021) and TACO (Li et al., 2023) established functional-correctness evaluation with tests. Synthetic datasets like KodCode (Xu et al., 2025) introduced a large-scale synthetic coding dataset with LLM spanning simple exercises to advanced algorithmic challenges. DELTA builds on this trend on a more fine-grained level, generating families of coding problems to isolate specific reasoning strategies and to test learnability and generalization under controlled distribution.

**Study on grokking.** Grokking (Power et al., 2022) is when a model memorizes small algorithmic training sets and only later suddenly generalizes after prolonged training. Explanations span train–test loss-landscape mismatch Liu et al. (2022), double-descent via pattern-learning speeds Davies et al. (2023), and gradient-spectrum splits between slow generalization and fast memorization Lee et al. (2024). Beyond traditional neural network settings, small transformers also grok on synthesized graph-based tasks (Wang et al., 2024; Abramov et al., 2025). Yet most work targets supervised, toy datasets; whether grokking occurs in RL on difficulty reasoning tasks remains unclear. To our knowledge, DELTA is the first to show that, under suitable training, grokking can emerge during RL fine-tuning of large language models.

## 6 DISCUSSION AND IMPLICATIONS FOR FUTURE STUDY

**A call to study the hard subset.** Recent benchmarks in math and code often report averages over mixed pools (Huan et al., 2025; Guha et al., 2025; Liu et al., 2025b;a), where a small subset of genuinely hard instances (those with pass@K = 0 for strong pretrained models) is washed out by easier items. Our results show that this "hard frontier" exhibits distinct learning dynamics—most notably a grokking-like phase transition under RL that can require hundreds or even thousands of training steps per problem. In large heterogeneous pools, the probability of repeatedly sampling and solving any single hard case is diluted, further suppressing the signal. We therefore advocate that future evaluations explicitly isolate and track this subset, so that progress on truly novel reasoning is not obscured by aggregate metrics.

**Beyond coding: from math to science.** Coding affords dense, verifiable feedback, which lets RL cross the learnability gap on problems that were previously unsolved. The same principle can extend to math and science when fine-grained signals are available: rubric-based scoring, stepwise checkers,

theorem-prover verification, and simulation- or constraint-based evaluators. We expect that porting the insights to these domains will enable RL to tackle currently unsolvable problems.

**How we train is as important as what we train.** While scaling data matters for LLM training, our results indicate that the training procedure is equally critical. First, a natural, curriculum-style progression does not always exist for all hard problems in a large mixed corpus; adding loosely related families does not reliably smooth learning and can fail to help (see Fig. 7). Second, concrete training choices—such as staged warm-up with dense rewards, experience replay, and verification/feedback-in-the-loop—show great promise for improving performance on hard-tier problems. More broadly, both *sharpening* (refining existing priors) and *discovery* (acquiring new strategies) can both happen; which regime emerges depends on the setup. The right RL infrastructure, reward design, data mixture, and level of task hardness can be decisive—together, these factors let us extract substantially more performance from RL and reach conclusions that would appear unattainable under different configurations.

## ACKNOWLEDGEMENTS

This work was supported in part by the National Science Foundation under the ACTION program. Computational resources were provided by the Allen Institute for AI (AI2) cluster. We thank all collaborators and reviewers for their valuable feedback and contributions.

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

# A   DATASET DETAILS

## A.1   MANUFACTORIA

*Manufactoria* is a classic Flash game (2010) in which players build automated factories to sort robots based on their colored tape patterns. The underlying logic resembles constructing finite-state automata or tag systems using two special node types (`puller`, `painter`). While the original game is implemented in 2D space, we re-formalize it into a custom programmatic syntax, as the syntax defined as a prompt below.

<hr>

**Prompt Template of Manufactoria Problems**

<hr>

# Manufactoria Solution DSL

A Domain Specific Language for describing Manufactoria puzzle solutions in text format.

## Overview

Manufactoria is a puzzle game where you build automated factories to sort robots based on their colored tape patterns. Robots enter your factory carrying sequences of colored tape, and you must route them to the correct destinations based on the given criteria.

## Game Mechanics

### Robots and Tape
- **Robots**: Each robot carries a sequence of colored tapes
- **Tape Colors**: Primary colors are Blue (B) and Red (R), with additional Yellow (Y) and Green (G) for advanced puzzles
- **Tape Representation**: Sequences are represented as strings (e.g., `RBRR`, `BBR`, or empty string `""`)

### Operations
- **Pull**: Remove tape from the front of the robot's sequence
- **Paint**: Add colored tape to the end of the robot's sequence
- **Route**: Direct robots through the factory based on their current tape state

### Objective
Route robots to the correct destinations based on their final tape configuration and the puzzle requirements:
- **Accepted**: Robot reaches the END node
- **Rejected**: Robot is routed to the NONE node, or caught in an infinite loop, or robot reaches the END node but fails to meet the puzzle's acceptance criteria

## DSL Syntax

### Program Structure

Every solution must start with a `START` directive and end with an `END` directive, wrapped in ```` ```manufactoria ...``` ````:

```manufactoria
START start:
    NEXT <next_node_id>

# Factory logic goes here
```

```
END end
```

### Node Types

#### 1. Puller Nodes

Pullers remove specific colors from the front of the robot's tape sequence and route based on the current front color.

**Red/Blue Puller:**

```manufactoria
PULLER_RB <node_id>:
    [R] <next_node_id>      # Route and remove color if front tape is Red
    [B] <next_node_id>      # Route and remove color if front tape is Blue
    [EMPTY] <next_node_id>  # Route if no tape or front tape is neither R nor B
```

**Yellow/Green Puller:**

```manufactoria
PULLER_YG <node_id>:
    [Y] <next_node_id>      # Route and remove color if front tape is Yellow
    [G] <next_node_id>      # Route and remove color if front tape is Green
    [EMPTY] <next_node_id>  # Route if no tape or front tape is neither Y nor G
```

**Note**: Unspecified branches default to `NONE`, which rejects the robot.

#### 2. Painter Nodes

Painters add colored tape to the end of the robot's sequence and continue to the next node.

```manufactoria
PAINTER_RED <node_id>:
    NEXT <next_node_id>

PAINTER_BLUE <node_id>:
    NEXT <next_node_id>

PAINTER_YELLOW <node_id>:
    NEXT <next_node_id>

PAINTER_GREEN <node_id>:
    NEXT <next_node_id>
```

## Syntax Rules

1. **Node IDs**: Must be unique identifiers (alphanumeric characters and underscores only)
2. **Comments**: Lines starting with `#` are comments (single-line only)
3. **Indentation**: Use consistent spaces or tabs for route definitions
4. **Case Sensitivity**: Colors must be uppercase (R, B, Y, G)
5. **Termination**:
   - Robots routed to `NONE` are rejected
   - Robots routed to the END node are accepted{objective_clause}

**6. \*\*Code Blocks\*\*:** Final factory code should be wrapped in triple
backticks with ``` markers

## Example

Here's a simple example that accepts robots with exactly one red tape
(ending tape should be empty):

```manufactoria
START start:
    NEXT entry

PULLER_RB entry:
    [R] end

END end
```

# Task
Your task is to design a factory with code with following functionality:

{criteria}

———————————— **The End of Prompt** ————————————

The criteria are defined in the Table 1 with different problem families.

| Problem Family | Difficulty | Criteria Examples |
|---|---|---|
| APPEND | BASIC | Accept any input and append the sequence RBR to the end of the tape. |
| EXACT | BASIC | Accept if the tape is exactly RBB. |
| START | BASIC | Accept if the tape starts with BR. |
| ENDS | EASY | Accept if the tape ends with BB. |
| REGEX | EASY | Accept if the tape matches the regex pattern (RBR)+(B)? exactly. |
| HAS | EASY | Accept if the tape contains the substring RYY (must be consecutive). |
| COMPR | EASY | Treat Blue as 1 and Red as 0. Accept if the binary number is greater than or equal to 13. |
| PREPEND | MEDIUM | Put BR at the beginning of the tape. |
| MUTATE | MEDIUM | Change all RB to BR sequentially. |
| BIT_OP | MEDIUM | Treat Blue as 1 and Red as 0. Apply bitwise OR with 16 to the binary number. |
| FDIV | HARD | Treat Blue as 1 and Red as 0. Apply floor division by 4 to the binary number. |
| SYMM | HARD | Accept strings that match the pattern $R\{n\}B\{n+1\}$ for any $n \geq 1$. |
| MINMAX | HARD | Treat Blue as 1 and Red as 0. Output the maximum of 11 and input. |
| ADD | HARD | Treat Blue as 1 and Red as 0. Apply add 8 to the binary number. |

Table 1: Manufactoria Problem families with difficulty levels and acceptance criteria.

We synthesize each problem family by starting from a parameterized template (alphabet, tape
operation(s), acceptance predicate, and any numeric thresholds) and then perturbing those parameters

within a constrained search space. Discrete knobs (e.g., start/end substrings, regex subpatterns, token rewrites) are toggled or swapped with near neighbors, while numeric knobs (e.g., bitwise constants, comparison thresholds, additive offsets, division factors) are jittered by small deltas drawn from a valid set (e.g., $\pm 1$, $\pm 2$, powers of two), with guards to keep the task well-posed and nontrivial.

A.1.1 Example Traces Comparison of a MANUFACTORIA-HAS Problem

Here is one example response of *Qwen3-4B-Instruct* on a problem in MANUFACTORIA-HAS: "Accept if the tape contains the substring BRRR (must be consecutive)".

───────────────── **Failure Case** ─────────────────

To design a **Manufactoria solution** that **accepts robots whose tape contains the substring 'BRRR'** (consecutive), and **rejects therwise**, we need to carefully build a factory that:

- **Detects** the presence of the sequence `BRRR` in the robot's tape.
- Only accepts robots when this exact pattern appears **somewhere** in their tape.
- Rejects all others (including those with `BRR`, `BRRR`, but not full `BRRR`, or other combinations).

---

## Understanding the Requirement

We are looking for **consecutive** `BRRR` | that is, the letters B followed by three R's in a row.

We can't just look for `B` and then three `R`s as separate steps | since the robot has a sequence, and we must detect **this exact substring**.

---

## Strategy

We will use **state-based pattern matching** using **pullers** to simulate a finite state machine.

We'll define states that track how close we are to matching "BRRR":

| State | Meaning |
|------|--------|
| `S0` | Start: looking for the first 'B' |
| `S1` | After seeing 'B': now expecting R |
| `S2` | After seeing 'R': now expecting R |
| `S3` | After seeing RR: now expecting R (final match) |
| `S4` | Match found: accept (go to END) |
| `S5` | Failed to match (hit any invalid character or didn't complete) → reject |

We'll use **PULLER_RB** nodes (since only R and B are used) to process the tape from front to back.

We want to **build a machine that scans** the robot's tape until it finds `BRRR`.

Once the substring is found, we route the robot to the **END**.

If it fails, we route it to a **rejection node** (via `NONE`).

---

## Design Steps

1. **Start** node sends to a scanning state.
2. Use a series of pullers to detect:
   - If the front is `B`, go to next state (`S1`)
   - If the front is `R`, stay in current state or fail.
   - If the tape is empty or neither, reject (fail).
3. After each `R`, advance state.
4. If we ever get `BRRR`, go to `END`.
5. If we see anything else (e.g., `G`, `Y`, or a non-R after `B`), reject.

> *Because the only tape colors in this puzzle are **B** and **R**, we **do not need** yellow or green.*
> *So we only need **PULLER_RB**.*

---

## Implementation

```manufactoria
START start:
    NEXT s0

s0:
    [B] s1
    [R] s5
    [EMPTY] s5

s1:
    [R] s2
    [B] s5
    [EMPTY] s5

s2:
    [R] s3
    [B] s5
    [EMPTY] s5

s3:
    [R] end
    [B] s5
    [EMPTY] s5

s5:
    NONE

END end
```

---

## Explanation

| State | Action |
|------|-------|
| `s0`: Start | Wait for `B` → go to `s1`. If `R` or empty → reject (`s5`). |
| `s1`: After 'B' | Expect R → go to `s2`. If `B` or empty → fail → reject. |

| `s2`: After 'RR' | Expect another R → go to `s3`. Any other → fail. |
| `s3`: After 'BRR' | If next is `R`, we have `BRRR`, go to **END**. Otherwise fail.
| `s5`: Failure state | All invalid or wrong transitions → `NONE` → **rejected** |

> **No alternative paths** are accepted.
> This correctly **only passes robots with "BRRR" in their tape**, and rejects:
> - `BRR` (missing last R)
> - `BBRR` (fails at `B` after `B`)
> - `BRBR` (breaks the sequence)
> - Any tape with `B` not followed by at least three `R`s
> - Empty or non-matching tapes

---

## Example Test Cases

| Input Tape | Accepted? | Why |
|----------|----------|-----|
| `BRRR` | Yes | Found `BRRR` → reaches `end` |
| `BBRRR` | Yes | First `B` → goes to `s1`, then `R` → `s2`, `R` → `s3`, `R` → `end` |
| `BRBR` | No | After `B`, then `R`, then `B` → fails (not `R`) → reject |
| `BRR` | No | No final `R` → rejected |
| `RRB` | No | No `B` at start → rejected |
| `` (empty) | No | Empty → rejects |
| `RBR` | No | No `B` at start, so never starts sequence → reject |

> Works as required.

---

## Final Solution

```manufactoria
START start:
    NEXT s0

s0:
    [B] s1
    [R] s5
    [EMPTY] s5

s1:
    [R] s2
    [B] s5
    [EMPTY] s5

s2:
    [R] s3
    [B] s5
    [EMPTY] s5

s3:
    [R] end
    [B] s5
    [EMPTY] s5

s5:
    NONE
```

```
END end
```

This solution correctly accepts robots with the substring **'BRRR'** and rejects all others.

──────────────────── **The End of Trace** ────────────────────

Here is another example response (partial) of *Qwen3-4B-Instruct* **after RL training** with grokking (as in Figure 6 (c)) on a test problem in MANUFACTORIA-HAS: "Accept if the tape contains the substring BRRR (must be consecutive)".

──────────────────── **Success Trace** ────────────────────

We are tasked with designing a factory in the **Manufactoria DSL** that:

- **Accepts** robots whose tape contains the substring **`BRRR`** (four consecutive characters: B, R, R, R)
- **Rejects** all others.

---

This is a **string matching** problem via finite automata. Since the tape is processed from left to right (front to back), and we are only allowed to use pullers (which remove from the front), we must simulate a finite state machine (FSM) that tracks progress toward detecting "BRRR".

We can't assume the input is long or ordered | so we have to detect the exact substring **"BRRR"** in the input sequence, **anywhere**, and accept if and only if it appears.

We must design an **automaton** that:

- Keeps track of how much of "BRRR" it has matched so far
- When it sees a match, it goes to an accepting state
- Otherwise, it continues scanning, possibly resetting or continuing partial matches

We do **not** allow adding tape (painting), only pulling (removing from front), and routing based on the current front character.

---

## Step 2: Define state transitions

We will build the automaton with states:

- `s0` → initial state
- `s1` → after matching B
- `s2` → after matching BR
- `s3` → after matching BRR
- `s4` → matched BRRR → ACCEPT

......

Now, what about the tape that is exactly `BRRR`?

- s0 → B → s1
- s1 → R → s2
- s2 → R → s3
- s3 → R → go to `end`

What about if the tape is `R`?

- s0 → R → s0 → then R → s0 → forever → never reaches end → rejected → correct.

What about `B`?

- s0 → B → s1 → then if next is R → s2 → etc.

So yes.

What about `BBR`?

- B → s1
- B → s1
- R → s2 → then if no more, it stops → never accepts → correct, because BRRR not present.

But what about `BBRRR`? → yes, after seeing second B → s1 → then R → s2 → R → s3 → R → end → Accepted

So it works.

Are there any **missing** cases?

What if the tape starts with Y?

- s0 → Y → goes to yg0 → Y → s0 → same → so eventually if B comes, we can proceed.

Eventually, if "BRRR" appears later, we will catch it.

What if "BRRR" appears in the middle? Like `X BRRR Y`?

- X → if R → s0 → then B → s1 → R → s2 → R → s3 → R → end

Perfect.

We never require the tape to start with B | just that BRRR appears anywhere.

So it's correct.

---

## Final Answer

```manufactoria
START start:
    NEXT s0

PULLER_RB s0:
    [R] s0
    [B] s1
```

```
    [EMPTY] yg0

PULLER_YG yg0:
    [Y] s0
    [G] s0
    [EMPTY] s0

PULLER_RB s1:
    [R] s2
    [B] s1
    [EMPTY] yg1

PULLER_YG yg1:
    [Y] s0
    [G] s0
    [EMPTY] s0

PULLER_RB s2:
    [R] s3
    [B] s1
    [EMPTY] yg2

PULLER_YG yg2:
    [Y] s0
    [G] s0
    [EMPTY] s0

PULLER_RB s3:
    [R] end
    [B] s1
    [EMPTY] yg3

PULLER_YG yg3:
    [Y] s0
    [G] s0
    [EMPTY] s0

END end
```

This factory accepts any robot whose tape contains the substring
`"BRRR"` (consecutive), and rejects otherwise.

This satisfies the requirement.

─────────────── **The End of Trace** ───────────────

## A.2 BOUNCINGSIM

We include a widely used community test—a 2D bouncing-ball simulation program—often treated as a proxy for geometry-aware reasoning in LLMs (Wiggers, 2025). The goal is to synthesize a program that simulates elastic collisions in polygonal containers and returns the exact object state at a queried timestamp; strong solutions require precise collision detection/response and numerically stable integration.

**Prompt design.** We provide a prompt example of the bouncing ball coding problems in `ROT_BOX` problem family below.

──────────────── **Prompt Template of BouncingSim Problems** ────────────────

## Polygon Dynamics Prediction
In this task, you will implement a single function predict_position(t)
that computes the 2D positions of all balls at an arbitrary future time
t under idealized mechanics. The function parses the scene configuration
(containers, balls, and physics/meta), reconstructs the motions, detects
and handles boundary collisions with finite-size treatment, and returns
a list where each element is the [x, y] position (rounded to 2 decimals)
of a ball at time t. Each evaluation of t must be computed directly from
initial conditions and scene mechanics with no hidden state or
accumulation across calls. Rendering, animation, and explanatory text
are out of scope; prefer closed-form reasoning and avoid coarse time-
stepping except where narrowly required for collision resolution.

### Mechanics (General)
- Kinematics: Use closed-form equations under constant acceleration:
x(t)=x0+vx0*t+0.5*ax*t^2, y(t)=y0+vy0*t+0.5*ay*t^2.
- Collisions: Perfectly elastic. Reflect velocity using v' = v -
2·dot(v, n^)·n^, where n^ is the inward unit normal at the contact.
- Finite size: Use polygon{polygon contact. Derive regular shapes from
('sides','radius','center','rotation'); irregular convex polygon balls
use provided vertices.
- Geometry: Irregular convex polygons (if present) are simple (non self-
intersecting). Ball finite size must be respected in all interactions.
- Units: Positions in meters; time in seconds; angles in radians;
velocities in m/s; accelerations in m/s^2.
- Cartesian Axes: +X is right, +Y is up.

### Constraints
- Implement only predict_position(t); no other entry points will be called.
- No global variables; no variables defined outside the function.
- Do not import external libraries (except math); do not perform I/O; do
not print; do not use randomness.
- Numerical output must be round(value, 2); normalize -0.0 to 0.0.

### Verification and output contract
- Return a list of positions per ball for the provided t: [[x1,y1],[x2,y2],...].
- Each call must be computed independently (no state carry-over between calls).
- You should assume that the ball will hit the wall and bounce back,
which will be verified in test cases.

### Scene description
#### Containers
- Container 1: regular polygon with 3 sides, radius 225.00m, center at
(750, 750); initial orientation 0.000 rad; constant angular velocity 0.170 rad/s

#### Objects

```
- Ball 1: regular polygon (3 sides), radius 40.0m, initial position
(750, 750), initial velocity (-220.61, 6.21) m/s
```

### Physics
```
- no effective gravity (treated as zero).
```

### Dynamics
```
- No additional time-varying mechanisms.
```

### Conventions for this scene
```
- Containers are convex regular polygons (parameters: 'sides', 'radius',
'center'), unless otherwise specified.
- Angle baseline: By default, the initial orientation is 0.000 rad,
pointing to the first vertex along +X (standard Cartesian axes);
positive angles rotate CCW about the container center.
- Polygon vertices (if provided) are CCW and form a simple convex polygon.
- Container 'radius' denotes the circumradius (meters).
- For balls: irregular convex polygons rely on provided vertices (no
radius mentioned); regular polygons may be derived from
'sides/radius/center/rotation'.
- Containers are kinematic (infinite mass, prescribed motion); impacts
do not alter container motion.
```

### Task
```
- Number of balls: 1
- Your should think step by step and write python code.
- The final output should be in the following format:
[Your thinking steps here ...](optional)
```python
[Your Python code here]
```
- Define predict_position(t) returning a list of length n_balls; each
element is [x_i, y_i] (rounded to 2 decimals) for Ball i at time t (seconds)
```

### Output
```
- Required format: function predict_position(t: float) -> [[x1,y1],
[x2,y2],...]; coordinates as 2-decimal floats
```

---
**The End of Prompt**
---

We construct a large-scale dataset for elastic collisions of polygonal objects in polygonal containers, designed to probe geometry-aware reasoning and numerically stable simulation in code-generating models (Wiggers, 2025). Each instance provides a fully specified physical scene and a programmatic task: predict the exact object state at one or more queried timestamps. Below we detail our scene taxonomy, generation and validation pipeline, prompt/evaluation protocol, and the difficulty schedule.

A.2.1 SCENE TAXONOMY

We factor the space of scenes into orthogonal "axes" that control distinct physical effects or composition, allowing systematic sampling and compositional generalization:

- ROT_OBJ (Inner rotation): the ball (modeled as a convex polygon) has nonzero angular velocity; collisions remain perfectly elastic.

- ROT_BOX (Outer rotation): the container rotates; optionally, time-varying angular speed is injected via a sinusoidal envelope.

- MOV_BOX (Outer translation): the container follows a prescribed path (sinusoidal or Lissajous), inducing moving-boundary reflections.

- GRAVITY: gravity can be tiny/small/large, tilted, or chaotic (random direction with time variation).

- `MULTI_BOX` (Multi-container): multiple non-overlapping polygonal containers are placed; a single ball is spawned in the first container unless otherwise specified.

- `MULTI_OBJ` (Multi-object): multiple balls are spawned in a single container with non-overlapping initial placement.

All containers and balls are convex polygons; collisions use a perfectly elastic model (restitution 1.0) with finite-size handling (ball centers are constrained by the container's incircle).

### A.2.2 Parameterization and Placement

Scenes are defined in a global, display-agnostic metric space. The workspace size is fixed to 1500 m × 1500 m with a baseline container diameter of 300 m. Difficulty scales the geometry (e.g., container diameter factor), polygon arity (number of sides), ball radii, speeds, and multiplicities. Objects are sampled and placed under strict feasibility constraints:

- Non-overlap: initial ball–ball overlap is rejected by a circle-approximation test; multi-container layouts must respect a minimum center-to-center gap.

- Feasible incircle: ball centers are sampled inside the container's incircle minus a safety margin; scenes violating this bound are rejected.

- Units: positions in meters; time in seconds; angles in radians; velocities and accelerations in SI units. All randomization is seeded and stored in scene metadata for reproducibility.

### A.2.3 Generation and Validation Pipeline

The dataset is produced in three stages, repeated for every requested problem family combination and difficulty level:

**(1) Scene synthesis.** Given a target problem family set (e.g., `ROT_BOX`) and difficulty, we draw parameters from problem-family-specific ranges (polygon arity, speeds, rotation rates, translation amplitudes, gravity modes) and write a normalized JSON scene: container(s), ball(s), physics (including time-varying profiles), and comprehensive metadata (difficulty name, seed, key timestamps, etc.). Difficulty levels scale geometry (container factor, polygon arity), ball radii, kinematics (linear and angular speeds), gravity complexity, and multiplicity (containers/balls) as shown in Table 2.

**(2) Numerical sanity check.** Each synthesized scene is validated for step-size stability before acceptance. We simulate the scene at a small set of reference timestamps under two integrators/time-steps (a validation baseline vs. the ground-truth step) and require the maximum screen-space deviation to remain below a tight threshold (15 px). Scenes that exceed this threshold or violate geometric feasibility (overlap or outside-incircle) are discarded and resampled up to a retry budget.

**(3) Dataset assembly.** For every accepted scene we choose evaluation timestamps and compute ground-truth positions using the higher-fidelity integrator. We then construct a task prompt and serialize a JSONL entry containing: messages (the task), a list of test assertions (per timestamp), the instance id, difficulty index, the explicit timestamp list, and an error tolerance tag (default 50px) used during automated checking.

### A.2.4 Splits and Composition

We design three complementary splits to probe distinct generalization properties. Each split is parameterized by which axes, difficulties, and timestamp regimes are exposed during training vs. evaluation.

**Design principles.** (1) Factorized skills. Axes isolate orthogonal mechanics (inner vs. outer rotation, moving boundaries, gravity, multiplicity, periodicity). (2) Controlled distribution shifts. Difficulty scales geometry, multiplicity, and dynamics; OOD splits increase complexity without changing the core mechanics.

Table 2: Problem-by-difficulty configurations (aggregated from generator defaults). Abbreviations: f = container diameter factor (relative to 300m base); out/in = outer/inner polygon sides; r = ball radius (m); v = linear speed range (m/s); $\omega$ = angular speed (rad/s); amp = translation amplitude (m); g = gravity mode; cts = number of boxes; n = number of balls.

| Problem family | Basic (0) | Easy (1) | Medium (2) | Hard (3) | Extreme (4) |
|---|---|---|---|---|---|
| ROT_OBJ | f 1.5; out 3–4; in 3–4; r 40; $\omega$ 0.1–0.2; v 200–400 | f 1.4; out 3–5; in 5–6; r 35; $\omega$ 0.2–0.5; v 400–600 | f 1.3; out 3–6; in 6–7; r 30; $\omega$ 0.5–1.0; v 600–800 | f 1.2; out 3–7; in 7–8; r 30; $\omega$ 1.0–2.0 (tv); v 600–800 | f 1.0; out 3–7; in 8; r 30; $\omega$ 2.0–2.5 (tv); v 600–800 |
| ROT_BOX | f 1.5; out 3–4; in 3–4; $\omega$ 0.1–0.2; v 200–400 | f 1.4; out 5–6; in 5–6; $\omega$ 0.2–0.5; v 400–600 | f 1.3; out 6–7; in 6–7; $\omega$ 0.5–1.0; v 600–800 | f 1.2; out 7–8; in 7–8; $\omega$ 1.0–1.5 (tv); v 800–1000 | f 0.8; out 8–10; in 8–10; $\omega$ 2.0–3.0 (tv); v 1000–1200 |
| MOV_BOX | f 1.5; out 3–4; amp 0–10; sin1d (0.1); v 200–400 | f 1.4; out 5–6; amp 20–40; sin1d (0.5); v 400–600 | f 1.3; out 6–7; amp 40–60; sin1d (1.0); v 600–800 | f 1.2; out 7–8; amp 60–90; Lissajous; v 800–1000 | f 1.0; out 8–10; amp 90–120; Lissajous (chaotic); v 1000–1200 |
| GRAVITY | f 1.5; out 3–4; g = tiny; v 200–400 | f 1.4; out 5–6; g = small; v 400–600 | f 1.3; out 6–7; g = large; v 600–800 | f 1.2; out 7–8; g = tilted; v 800–1000 | f 1.0; out 8–10; g = tilted; v 1000–1200 |
| MULTI_BOX | cts 2; f 1.5; out 3–4; r 40; v 200–400 | cts 2; f 1.4; out 5–6; r 35; v 400–600 | cts 3; f 1.3; out 6–7; r 30; v 600–800 | cts 4; f 1.2; out 7–8; r 25; v 800–1000 | cts 6; f 1.0; out 8–10; r 20; v 1000–1200 |
| ROT_BALL | n 2; f 2.5; out 3–6; in 3–6; r 20; v 200–400 | n 3; f 2.5; out 3–6; r 20; v 400–600 | n 4–5; f 2.5; out 3–6; r 20; v 600–800 | n 5–6; f 2.5; out 3–6; r 20; v 800–1000 | n 7–9; f 2.5; out 3–6; r 20; v 1000–1200 |

**Explorative generalization (within-family difficulty shift).** This split tests robustness to increased geometric/dynamic complexity while keeping the same "skill". We train on single-family scenes at Basic difficulty and evaluate on the same family at higher difficulties.

- Train: single-family scenes at Basic (0). We generate 1000 examples in such a split.
- Test (ID): single-family scenes at Basic (0). We generate 100 additional examples in such a split.
- Test (OOD): Easy–Extreme (1–4) at the same family; We generate 100 additional examples in each difficulty.
- Rationale: isolates the effect of tighter geometry (smaller containers, more sides), higher velocities, stronger/tilted gravity, and larger multiplicity (more containers/balls), while holding the family-specific mechanics fixed.

**Compositional generalization (skill composition).** This split probes whether models learned modular skills that compose. Concretely, we exemplify by composing inner and outer rotations at test time after training on them in isolation.

- Train: ROTAT_BOX (outer rotation only) and ROTAT_OBJ (inner rotation only), both at Basic difficulty. We generate 1000 examples in each family.
- Test (OOD composition): ROTAT_BOX_OBJ = (outer+inner rotation simultaneously) at Basic (0) level. Container angular velocity and object spin are drawn independently at the current difficulty level. We generate 100 additional examples in such a split.
- Rationale: assesses whether learned collision handling in a rotating frame combines with inner-spin kinematics without interference.

**Transformative generalization (qualitative strategy change).** Here the test-time data is qualitatively different from anything seen in training—for instance, perfectly periodic trajectories that arise from special construction.

- Train: single-family scenes at Basic (0). We generate 1000 examples in such a split.
- Test (transformative OOD): periodic configurations (even-sided container, symmetry-aligned initial velocity) using list-prompt mode with a fixed periodic grid; we evaluate cycle consistency and phase accuracy over evenly spaced timestamps. We provide an example theorem below that supports such a periodic case construction.
- Rationale: measures whether models trained on generic dynamics can extrapolate to a qualitatively different but mathematically structured regime (near-1D periodic motion in polygonal symmetry).

**Periodic Construction (transformative setting).**    We exploit a closed-form condition that yields perfectly periodic, normal "shuttle" trajectories between two concentric, co-rotating regular polygons. This result underpins the periodic test cases in our ROT_BOX transformative split and provides an analytical knob to dial the fundamental period via the angular velocity.

**Theorem 1** (Periodic bounce between two concentric regular $n$-gons). ***Setup.*** *Let $P_o$ and $P_i$ be two concentric regular $n$-gons ($n \geq 3$) with circumradii $R_o > R_i > 0$. Both polygons rotate rigidly with the same constant angular velocity $\omega$ about their common center. At time $t = 0$ a point mass ("ball") is placed on the inward normal to a side of $P_o$ and moves with speed $v > 0$ along that normal toward $P_i$. Collisions with sides are perfectly elastic, and motion is confined to the annular region between the polygons. The initial pose has one vertex on the $+x$-axis.*

*Let $a(R) := R\cos(\pi/n)$ denote the apothem of a regular $n$-gon with circumradius $R$, and define the normal gap*

$$\Delta := a(R_o) - a(R_i) = (R_o - R_i)\cos\left(\tfrac{\pi}{n}\right).$$

*Thus $\Delta$ is the (signed) distance between the parallel supporting lines of the corresponding side family in $P_o$ and $P_i$.*

***Claim (closed-form condition).*** *The ball executes uniform periodic motion—bouncing back and forth at constant speed along a fixed set of parallel sides with a repeating impact pattern—if and only if there exists an integer $k \in \mathbb{Z}$ such that*

$$\boxed{\omega = \frac{k \cdot 2\pi v}{n(R_o - R_i)\cos\left(\tfrac{\pi}{n}\right)}}$$

*Equivalently, with the one-way flight time*

$$t_{\text{fly}} = \frac{\Delta}{v} = \frac{(R_o - R_i)\cos(\pi/n)}{v},$$

*the periodicity condition is*

$$\boxed{\omega\, t_{\text{fly}} = k \cdot \frac{2\pi}{n}}.$$

*When this holds, the fundamental bounce period and the orientation recurrence are*

$$T_{\text{bounce}} = 2\, t_{\text{fly}} = \frac{2(R_o - R_i)\cos(\pi/n)}{v}, \qquad T_{\text{orient}} = \frac{2\pi}{|\omega|} = \frac{n\Delta}{|k|v}.$$

*The minimal nonzero periodic rotation corresponds to $|k| = 1$.*

***Proof sketch.*** *(1) In a regular $n$-gon, opposite sides are parallel; the distance between their supporting lines is $2a(R)$. For concentric, co-oriented $P_o, P_i$, the normal gap between the corresponding supports is $\Delta = a(R_o) - a(R_i)$. (2) Launching exactly along a side normal produces specular reflections that preserve a straight, normal shuttle between parallel side families; the speed remains $v$, so each one-way flight takes $t_{\text{fly}} = \Delta/v$. (3) During a one-way flight, the polygons rotate by $\omega\, t_{\text{fly}}$. For the next impact to occur on a side parallel to the previous one (so that the normal shuttle and impact geometry repeat), the side orientations must recur, which in a regular $n$-gon happens modulo $2\pi/n$. Hence $\omega\, t_{\text{fly}} \equiv 0 \pmod{2\pi/n}$, yielding the stated condition.*

**Construction recipe for ROT_BOX (periodic).**    To instantiate periodic test scenes in the transformative split

1. Choose $n$ (even $n$ makes the normal families align with diameters) and set circumradii $(R_o, R_i)$ (or effective radii after finite-size shrink/expand).

2. Pick a speed $v > 0$ and launch along a side normal of $P_o$ (avoid vertex alignment by a tiny phase offset).

3. Set the box angular velocity using $|k| = 1$ in the closed form, $\omega \leftarrow \frac{2\pi v}{n,(R_o - R_i)\cos(\pi/n)}$, and co-rotate any inner boundary if present, or equivalently use $\omega_{\text{rel}}$ for differential rotations.

4. The resulting shuttle has $T_{\text{bounce}} = 2(R_o - R_i)\cos(\pi/n)/v$ and repeats in orientation every $T_{\text{orient}} = n\Delta/v$. For evaluation, sample timestamps on a uniform grid over several bounce periods to probe phase stability.

### A.3 COMPETITION CODING

*Competition Code* is a well-established domain where participants solve complex algorithmic problems. For a specified problem, the solver program is required to generate the correct output for every input in the provided test suite. We curate 5 algorithmic families and collect several problems per family from various well-known competitive programming platforms. We propose a phased perturbation pipeline to create a comprehensive OOD dataset.

#### A.3.1 SEED FAMILIES AND COVERAGE

We curate 3-5 seeds per algorithmic family. The current collection includes:

- **Mo's Algorithm (4):** *LuoguP1494, LuoguP4462, LuoguP4887, LuoguP5047*

- **Segment Tree Decomposition (3)**: *CF981E, CF1140F, LuoguP5787*

- **CDQ D&C (3):** *CF848C, CF1045G, LuoguP4093*.

- **Meet-in-the-Middle**: *CEOI2015-D2T1, LuoguP2962, SPOJ-ABCDEF, USACO2012USOpen-GoldP3*

- **Square Root Decomposition (5):** *CF710D, CF797E, CF1207F, LuoguP3396, LuoguP8250*.

Each seed problem is tagged with public problem code in websites like *CodeForces*, *AtCoder*, and *Luogu*. Per seed, we target *5-10 perturbation strategies* (configurable; default 10). For narrative coverage, we maintain a library of *20 background templates* (e.g., *Campus Life*, *Ancient Warfare*, *Cyber Security*, *Energy Grid*, *Xuanhuan Fantasy*), and by default rewrite each perturbed seed into all backgrounds.

#### A.3.2 SYNTHESIS PIPELINE

**Phase 1: Standardize seed problems.** This phase transforms heterogeneous problem statements into a unified specification. First, the framework parses raw Markdown to extract core fields such as the problem statement, input/output formats, constraints, and examples, and utilize LLMs to reduce typographic ambiguities and make semantic clarifications.

**Phase 2: Produce enumeration-based solutions for standardized seed problems.** This phase generates a diverse set of feasible, though not necessarily optimal, reference implementations for each standardized seed problem. Emphasis is placed on reliability rather than optimality, ensuring we have correct solutions for small test cases.

**Phase 3: Produce enumeration-based test case generators for standardized seed problems.** This phase synthesizes test case generators grounded in original seed problems. By curating prompts for LLMs, generators are designed to cover representative distributions and adversarial conditions.

**Phase 4: Generate perturbation strategies.** This phase generates strategies how to perturb problems systematically. Each strategy seed is curated by a human expert with at least 8 years of competitive programming experience and designed for making a perturbation while keep the main solution unchanged. These strategy seeds are standardized and extended to strategies with detailed instructions.

**Phase 5-7: Generate perturbed problems, enumeration-based solutions and test case generators according to strategies.** Phase 5 generates standardized perturbed problem statements, based on perturbation strategies. Similar to phase 2 and phase 3, we generate corresponding solutions and test case generators based on enumeration. When generating solutions, we provide the original problem and solution to effectively improve the reliability.

**Phase 8: Produce input constraint sanity check test case generators for standardized perturbed problems.** To enhance the robustness of our evaluation, this phase produces input constraint sanity check test case generators. Curated test case generators are designed for testing whether the solution code can handle big test cases in a reasonable small time. Test case constraints are manually adapted to the Python programming setting, guaranteeing no brute-force solutions can pass and all correct Python solutions can be accepted.

**Phase 9: Produce background rewrites.** Finally, this phase provides an effective approach to generate OOD samples. By utilizing 20 background settings, the standardized perturbed seed problems are rewritten in different background stories, maintaining the same input/output formats and solutions. All these rewritten problems are final and ready to be involved in training.

### A.3.3 EXAMPLE 1: SEGMENT TREE DECOMPOSITION – BIPARTITE OVER TIME

**Seed (excerpt).**

> "Given (n, m, k). Each of the (m) edges is active on an interval ([l, r]) over the discrete timeline (1..k). For each time (t), determine whether the active subgraph is bipartite."

**Perturbation strategies (from Phase 2, sample).**

- **Two-interval activation.** Replace each edge's interval ([l, r]) with exactly two disjoint subintervals ($[l_1, r_1], [l_2, r_2]$). The solver continues to use DSU-rollback over a segment tree covering time.
- **Interval→Event rewrite.** Convert each interval to two explicit events: an add at (l), a remove at (r+1). Feed the event list unchanged into the segment-tree over time.
- **Event-pair splitting.** Expand each add/remove into two sub-events (e.g., *prepare/apply*) to stress timeline density without changing the rollback design.

**Before/After (Strategy-level variant).** *Before (seed):* time-varying edges with single intervals ([l, r]). *After (strategy 1): "Each edge is active exactly on two disjoint intervals ($[l_1, r_1]$) and ($[l_2, r_2]$). For each (t) in (1..k), is the subgraph bipartite?"* Algorithmic essence and complexity remain the same: DSU with rollback over a segment tree on the time axis, $O((n + m) \log k)$.

### A.3.4 EXAMPLE 2: SQUARE ROOT DECOMPOSITION – HASH-BUCKET GROUP SUMS

**Seed (excerpt).**

> "Given an array value. For many queries with modulus (p¡n), report the sum of numbers in bucket (x), where index (k) belongs to bucket ($k \mod p$). Updates assign value$_i \leftarrow y$."

**Strategy-level perturbation (background-agnostic).** *Before:* group by ($k \mod p$). *After:* **Grouped Sequence Sum and Update Queries:**

> "Define $H(i) = \sum_{k=0}^{K-1} S_k i^k \mod M$. Sum queries ask for the total over indices mapping to a given hash value (g); updates set $A_i \leftarrow x$."

This preserves the bucket-sum structure and the $O(\cdot)$ behavior under small-(M) caching and updates, matching the seed's enumeration profile while modestly changing the grouping function.

**Background rewrite (Campus Life).** *Before (strategy-level):* abstract group sums under (H(i)). *After (background):* **Campus Club Scores:**

> "Student IDs (1..N) are assigned to clubs by a polynomial function (C(i)). Queries ask for the total score in club (g); updates change a student's score."

Narrative terms shift (students/clubs/scores), but the formal mapping (C(i)) and the I/O grammar remain intact so the variant's enumerator and the background rewrite both agree on the 100-case oracle.

### A.3.5 SUMMARY

By enumeration-first solutions and enforcing strategy-level clarity before rewriting, the pipeline makes large-scale, verifiable perturbation feasible. Standardization, deterministic test generation, and background consistency checks together ensure that every variant—despite narrative diversity—remains faithful to the core algorithm and produces outputs consistent with the seed's brute-force oracle. This methodology yields rich, well-structured families suitable for training, evaluation, and pedagogical use.

## B  EXPERIMENT DETAILS

**Models.** We use *Qwen3-4B-Instruct* as the reference instruction-tuned model for all experiments in this paper.

**Training Details.** We fine-tune with GRPO (Guo et al., 2025) using the Open-Instruct framework[3]. Unless otherwise noted, the key arguments are:

```
--beta 0.0 \
--num_unique_prompts_rollout 48 \
--num_samples_per_prompt_rollout 16 \
--kl_estimator kl3 \
--learning_rate 5e-7 \
--max_token_length 12240 \
--max_prompt_token_length 2048 \
--response_length 10192 \
--pack_length 12240 \
--apply_verifiable_reward true \
--non_stop_penalty True \
--non_stop_penalty_value 0.0 \
--temperature 1.0 \
--total_episodes 1000000 \
--deepspeed_stage 2 \
--per_device_train_batch_size 1 \
--num_mini_batches 1 \
--num_learners_per_node 8 \
--num_epochs 1 \
--vllm_tensor_parallel_size 1 \
--clip_higher 0.3 \
--vllm_num_engines 8 \
--lr_scheduler_type constant \
--seed 1 \
--gradient_checkpointing \
```

Across all experiments—including the multi-stage schedules in the paper—we vary only (i) the train/eval datasets, (ii) the base/reference model, and (iii) the scoring mode (full-pass reward vs. per-test reward) to match the setting.

**Datasets for learnability (Section 3).** `Manufactoria-HAS`: 742 training and 100 test examples. `Manufactoria-START/APPEND/EXACT`: 350 training examples in total across the three families. `Manufactoria-REGEX`: 560 training examples. `Manufactoria-COMPR`: 535 training examples.

**Datasets for generalization (Section 4).** Unless otherwise specified, for each curated problem family and each difficulty, we sample 1,000 training problems (Appendix A.2.4). In the setup of Figure 9(a), the training set contains six families at the *Basic* level, totaling 6,000 training samples. Evaluation comprises:

- **In-distribution (ID):** 100 test samples from the same *Basic* difficulty as training.
- **Explorative (OOD):** 100 test samples per family at each higher difficulty (*Easy*, *Medium*, *Hard*).
- **Compositional (OOD):** 100 test samples per composed family at *Basic* difficulty.
- **Transformational (OOD):** 100 test samples per setting.

**Evaluation Protocol.** Evaluation uses the same sampling configuration as training. Each score is averaged over 4 runs.

**Compute Resources.** Each RL run uses 16 NVIDIA H100 GPUs across two nodes and completes in ∼3 days for 1,000 optimization steps.

---

[3]https://github.com/allenai/open-instruct

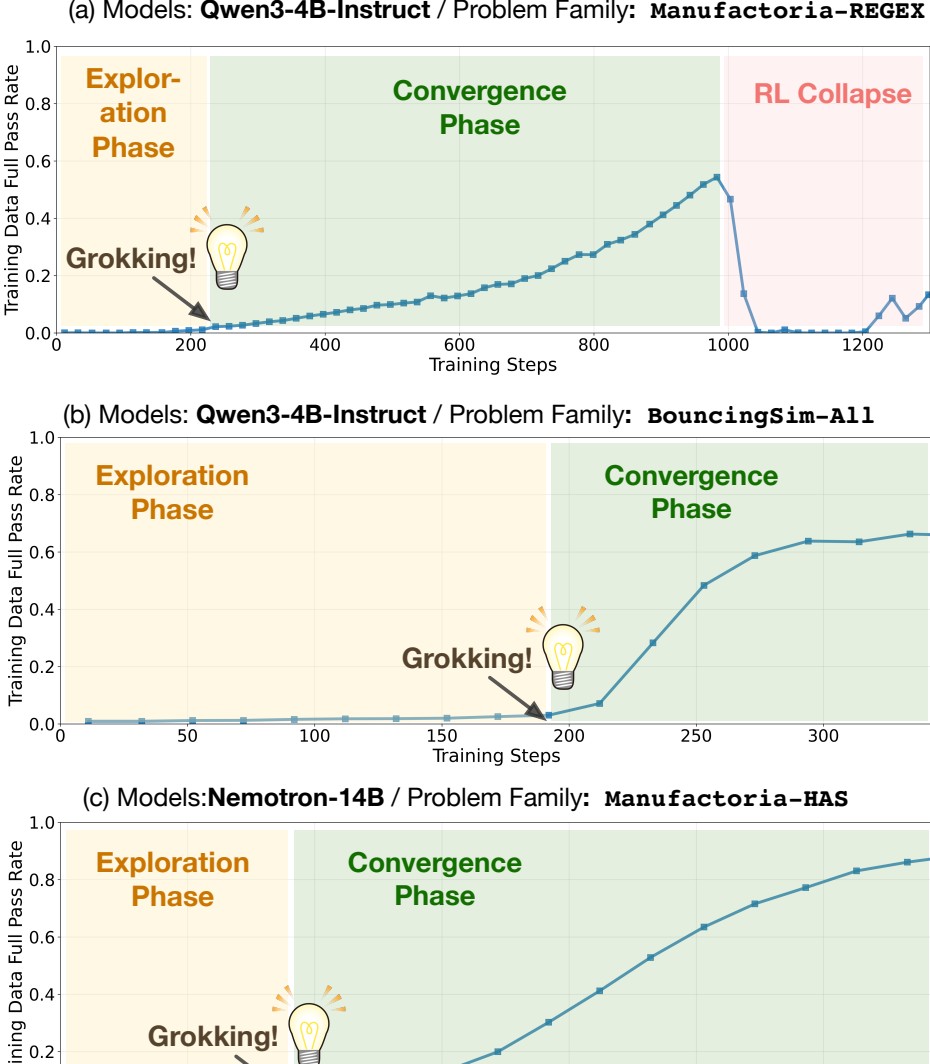

Figure 10: **Grokking across models and tasks.** (a) *Qwen3-4B-Instruct* on *Manufactoria–REGEX*; (b) *Qwen3-4B-Instruct* on *BouncingSim–All* (same training setup as in Figure 9); (c) *Nemotron-14B* on *Manufactoria–HAS*. Curves plot *training-data full pass rate* versus training steps. A consistent pattern emerges: a long exploration phase, an abrupt grokking transition, and a convergence regime; (a) also exhibits an RL collapse when training continues past convergence.

## C ADDITIONAL EXPERIMENTS

### C.1 GROKKING GENERALIZES ACROSS MODELS AND PROBLEM FAMILIES

Figure 10 demonstrates that the *RL grokking* phenomenon, an extended low-signal exploration phase followed by an abrupt phase transition and rapid convergence in training-data full-pass rate, can arise across (i) model sizes and families and (ii) distinct problem scopes.

Panel (a) shows *Qwen3-4B-Instruct* trained on `Manufactoria-REGEX`. After a long plateau, performance surges and subsequently enters a convergence regime. Continued training eventually triggers an *RL collapse*, highlighting the need for stabilization or early stopping once solutions consolidate. Panel (b) uses the same model on *BouncingSim–All*, a real-world ball-bouncing simulation

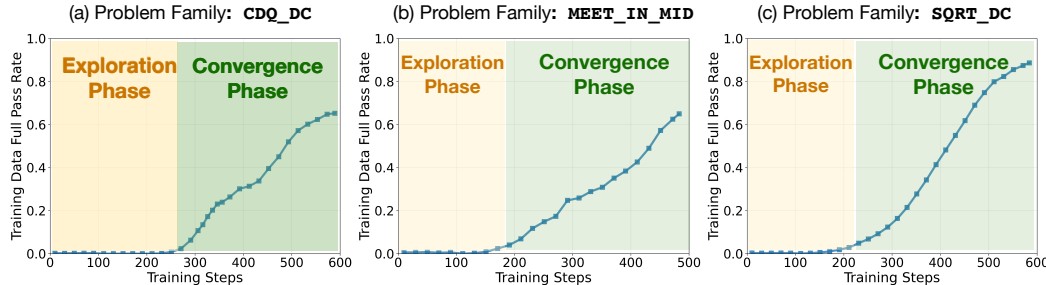

Figure 11: **Grokking across competition code tasks.** (a) *CDQ-DC*; (b) *MEET-IN-MID*; (c) *SQRT-DC*. Curves plot training data full pass rate versus training steps.

coding suite for real-world coding tasks. The same exploration to phase-transition to a convergence pattern appears. Panel (c) swaps the model family and scale to **Nemotron-14B** on *Manufactoria–HAS*, again reproducing the grokking phenomenon. We also provide additional experiments on the competition coding task problem families in Figure 11 that demonstrate the same trend.

Together, these results indicate that grokking is not an artifact of a particular backbone or a single synthetic family. It emerges with different parameter counts, across independent model lineages, and on tasks ranging from symbolic program synthesis to physics-driven simulation code. This supports the view that RL can *discover* new procedural strategies rather than merely sharpening pre-trained ones.

## C.2 WARM-UP BENEFITS BEYOND THE "PASS@K=0" PROBLEMS

Warm-up with per-test rewards is not only a rescue mechanism for tasks where the base policy never succeeds; it also helps when the initial success probability is small but non-zero ($pass@k = \epsilon > 0$). In this regime the binary full-pass reward still provides a weak and high-variance signal, which can lead to slow or unstable improvement. A short warm-up phase with dense, per-test rewards (here: 100 steps) (i) accelerates discovery of partially correct behaviors, (ii) better stability, and (iii) delivers a more reliable starting point for the subsequent binary-reward phase. Empirically, we observe faster and steadier convergence with warm-up, whereas training that optimizes full-pass from scratch can remain sluggish and brittle, sometimes exhibiting late-stage regressions even after partial progress.

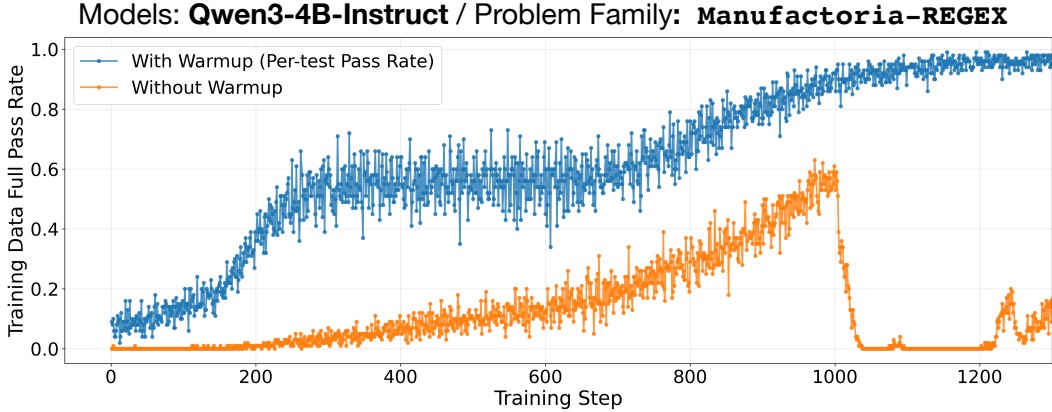

Figure 12: **Warm-up helps when *pass@k* is small but non-zero.** Training curves on *Manufactoria–REGEX* with *Qwen3-4B-Instruct*. The blue curve is trained **after a 100-step warm-up** using per-test rewards, then switched to the binary full-pass objective; it achieves faster and steadier gains. The orange curve trains full-pass from scratch and improves slowly with occasional regressions.

