# OpenReview forum: "RL Grokking Recipe: How Does RL Unlock and Transfer New Algorithms in LLMs?"
_ICLR.cc/2026/Conference — ICLR 2026 Poster_

### Official Review · Reviewer_wA9H · 2025-10-21

**Soundness:** 3
**Presentation:** 3
**Contribution:** 3
**Rating:** 8
**Confidence:** 4

**Summary:**

The authors of this paper present a controlled, fully verifiable benchmark suite for program synthesis spanning Manufactoria DSL puzzles, BouncingSim bouncing ball simulation, and competitive programming tasks. The authors position this benchmark suite as a programming-domain counterpart of OMEGA (synthesizable math families), producing unit-test verifiable instances, yielding rich per-test reward signals and making external tool-shortcutting much harder.

With this benchmark suite, the authors investigate learnability, whether RL can train a model to solve problem families that the base model fails at, even after many sampled attempts (high pass@K), and generalization, whether the learned procedures transfer to OOD problems along the exploratory, compositional, and transformative axes.

The key empirical finding is that a two-phase reward schedule, which involves a warm-up stage with dense per-test rewards and then switching to a binary full-pass reward, induces a grokking-like transition. After a long, low-reward plateau, performance abruptly jumps to high accuracy and then stabilizes, including on families where the base model had a pass@K=0. Across tasks, gains transfer well on the exploratory and compositional splits, while performance on the transformative split remains weak. The authors interpret this as evidence that RL can unlock solution strategies beyond what is reachable by sampling the base model alone. The authors also caution that the warm-up is not a universal recipe and the benefits depend on model capacity, target family difficulty, and curriculum compatibility.

**Strengths:**

**Clear Learnability Results**: On a family where the base model has pass@K=0, the two-phase (dense to binary) schedule yields a grokking-like transition from failure to full pass. The grokking phase is a strong signal that the model has found an algorithmic procedure.

**Generalization Along Principled Axes**: Using OMEGA-style exploratory, compositional, and transformative splits, they show strong transfer on the first two and pinpoint limits on the third. This is a useful signal for where future work is needed.

**Comprehensiveness**: Broad experimental sweep across domains, families, defining thoughtful difficulty ladders, and training recipes, including ablations on replay, feedback-in-the-loop, and curriculum-based data, supporting the central claims.

**Dataset Contribution**: The authors supply an OOD problem space by introducing a bespoke Manufactoria DSL, a new syntax unlikely to appear in pretraining and pairing it with templated generators. The authors also organize families into a scalable difficulty ladder (Basic, Easy, Medium, Hard) based on aggregate model performance, so researchers can stage experiments by capability.

**Weaknesses:**

**Under Characterized Reward Switch Schedule:**
The two-staged dense -> binary schedule is central to the learnability claim, yet the paper does not ablate warm-up length or switch timing. The only concrete examples are an illustrative 100-step saturation during per-test warm-up, 450-step post-switch exploration, and a 100-step warm-up case in the appendix. The schedule hyperparameters otherwise appear fixed. This leaves open the minimal warm-up needed, whether earlier/later switches change sample efficiency or stability, and the robustness of this claim. More details strengthen the result and clarify robustness.

**Evidence for Curriculum Task Compatibility:**
The paper’s claim that “curriculum must be task-compatible” is compelling, but with only two examples (REGEX -> HAS, COMPR -> HAS), it is hard to generalize and see if the effect holds across other families or difficulty tiers, or whether there are cases of strong negative transfer. A small source x target compatibility/staged results matrix (warm-up on source S -> RL on target T) across more families/tiers, alongside this paper’s BouncingSim slice, would strengthen the paper and enable the research community to understand and quantify this paper’s claims of how curriculum design impacts learnability dynamics.

**Questions:**

Could the authors clarify how they decided to switch from dense to binary (fixed step vs plateau/success trigger), the warm-up length (was it only illustrative?), and whether they tried a gradual weight shift from dense to binary?

Minor Comments/Typos:
- Specicaly -> Specifically
- consruction -> construction
- BouncingSim vs BounceSim. Figure 1 uses BounceSim while the rest of the paper uses BouncingSim. Is this on purpose?
- Figure 1 claims four axes, including “Domain level”, but experiments report three (exploratory, compositional, and transformative). Is there a reason why there were no results from the Domain-level?

---

> ### Author Response · Authors · 2025-11-20
>
> We thank the reviewer for the constructive and valuable feedback.
>
> ### **Reward Switch Schedule**
>
> Thank you for raising this important point. The purpose of the dense-reward warm-up phase is simply to push RLVR out of the absolute zero region for the full pass rate. Once the model achieves a small but non-zero full-pass accuracy (e.g., after ~50 steps in many settings), standard GRPO begins to make progress, and we observed that switching at 100 vs. 200 steps makes minimal difference which both reliably lead to the grokking phenomenon.
>
> In practice, this procedure can be fully automated: one can set a minimal full-pass threshold ε, and end the warm-up phase immediately once this threshold is exceeded. In our paper, we used a fixed checkpoint for clarity of illustration rather than an adaptive threshold, but we agree this is a natural extension.
>
> ### **Curriculum Task Compatibility**
>
> We appreciate this suggestion. While a full *N × N* curriculum compatibility grid is computationally expensive since each RL stage requires 2 × 8 H100 GPUs for two days, so a complete grid across *N* families would cost approximately
> $
> 2\text{ (stages)} \times N^2 \times 16\text{ (GPUs)} \times 50\text{ GPU-hours} \times $2.5/\text{hr}
> = $4000 \times N^2,
> $
>
> We agree that a partial matrix can provide useful insight. Due to the time constraint of the rebuttal period, we computed a 4 × 4 compatibility matrix on the four *easy-tier* tasks. For each source→target pair, we ran stage 2/3 with 1000 RL steps each; if the final accuracy on the target exceeded 60%, we marked the cell as Y, otherwise N. The results indicate that transfer is highly selective:
>
> |           | ENDS | REGEX | HAS | COMPR |
> | --------- | ---- | ----- | --- | ----- |
> | **ENDS**  | \    | N     | N   | N     |
> | **REGEX** | N    | \     | Y   | N     |
> | **HAS**   | N    | Y     | \   | N     |
> | **COMPR** | N    | N     | N   | \     |
>
>
> ### **Other**
>
> We thank the reviewer for the careful reading. We have addressed all detailed comments in the revised manuscript. Due to the high computational cost of full domain-level generalization grids (again N² experiments), we did not run them end-to-end; however, we included an illustrative figure to highlight that our dataset design makes such controlled evaluations possible and can support future work in this direction.

---

> > ### Comment · Reviewer_wA9H · 2025-11-27
> >
> > I thank the authors for their detailed response and rebuttal.
> >
> > I appreciate the additional clarification that the dense to binary reward switch can, in principle, be automated by triggering once performance exceeds some threshold ε. My concern is less about the existence of a useful switch and more about the impact and sensitivity of different schedules across tasks. Since this mechanism is a central claim of the paper, it would be stronger with an explicit robust experiment characterizing how the results depend on the choice of warm up length and switch criterion (or how it doesn't).
> >
> > Regarding the curriculum task compatibility, the resulting matrix shows that only REGEX and HAS exhibit positive transfer, while other pairs do not, which nicely supports the claim that curriculum benefits are highly selective rather than generic.
> >
> > Given the clarifications and additional experiments, I maintain my positive overall assessment and my original score.

---

### Official Review · Reviewer_eAUM · 2025-11-01

**Soundness:** 3
**Presentation:** 3
**Contribution:** 4
**Rating:** 8
**Confidence:** 3

**Summary:**

The paper examines the learnability and generalization of the RL algorithm in the domain of large language models. The paper presents two benchmarks, Manufactoria and BouncingSim, for testing novel algorithmic skills and compositional algorithmic skills. These tasks are proven to be unsolvable for LLMs with an adequate sample size. To test the learnability of the collected benchmark, the authors design a method to assign partial credit to a single test case and then transition to full credit. With this method, the RL algorithm (GRPO) exhibits the grokking phenomenon, and its performance grows rapidly within a few steps. Additionally, the experiment suggests that with the selected curriculum, the RL algorithm can learn more effectively. Bouncing2D is designed to test the generalization. The experiment results show that RL can exhibit good exploration and compositional generalization, but still yield poor transformative results.

**Strengths:**

1. Two different benchmarks to test the learnability and generalization of RL algorithms.
2. There are debates about the Grokking phenomenon; the work provides a testbed for testing Grokking, which is quite hard for the most performing language model, GPT-5. With a simple warm-up that includes unit-test credit assignment, the model exhibits a 'grokking' moment in the experiment.

**Weaknesses:**

1. The justification of OOD-ness is not trivial to me, see Questions 1 and 2 below.
2. The paper discusses learnability and generalization on two different datasets. There is no strong correlation between these two datasets. See Questions 4 and 5 below.

**Questions:**

1. In section 2.1, these problems are not common programming problems or Turing-machine tasks. What do the problems belong to in terms of problem complexity? Does the problem belong to push-down automata or simply a finite-state machine? Or is the problem much harder than a Turing machine?
2. Following Q1, how much OOD-ness comes from the program syntax? If the program syntax is written in Python, can modern LLMs solve these problem families? Does the Grokking phenomenon come from the syntax? Does the language model suddenly understand the syntax and solve the problem perfectly?
3. In section 3.2 and Figure 6, how does the result `its convergence speed is still slower than the baseline GRPO algorithm, likely because the reused traces are off-policy` come from? Figure 6 shows Experience Replay and baseline converge to a common performance.
4. Although Bouncing2D is designed for testing generalization, what is the learnability of different difficult tiers?
5. What kinds of generalization ability are needed to learn across different families in Manufactoria? Or what are the atomic algorithmic skills needed between families to make the generalization impossible?

---

> ### Author Response · Authors · 2025-11-20
>
> We thank the reviewer for the constructive and valuable feedback.
>
>
> ### **Problem Complexity of Manufactoria**
>
> The 2010 Manufactoria puzzles can be modeled as a finite-control device equipped with one unbounded FIFO tape, which is Turing-equivalent. However, solving problems in Manufactoria is substantially harder than operating a standard Turing machine: the read head can access only the left boundary of the tape, and the write head only the right boundary, removing the ability to freely navigate or overwrite arbitrary tape locations. As a result, implementing algorithmic logic requires weaving data through constrained pathways, making the effective programming model considerably more restrictive and intricate than classical Turing-style computation.
>
>
> ### **Problem Syntax of Manufactoria**
>
> We appreciate the reviewer’s question about how much difficulty arises from syntax versus underlying algorithmic reasoning. Figure 2 can be interpreted as an ablation along this axis: modern LLMs solve the “easy” tier almost perfectly, demonstrating strong mastery of the new Manufactoria syntax. As tasks become more algorithmically complex, performance drops even though the models continue to produce syntactically correct solutions. Detailed traces in Appendix A further illustrate this: in failure cases, the logic is wrong even when the syntax is essentially flawless. These observations suggest that the challenge and the grokking behaviour stem primarily from the underlying algorithmic structure rather than from learning the syntax.
>
> ### **Convergence Speed of Figure 6**
>
> Although the orange and green curves (experience replay variants) reach the grokking onset earlier, their subsequent improvement is slower than that of the blue GRPO baseline. Our statement about slower “convergence speed” refers specifically to the post-grokking slope: the orange/green curves exhibit lower slope ratios after the turning point, even though the end performance converges to a similar level.
>
> ### **Generalization Analysis of Manufatoria**
>
> Manufactoria offers a rich testbed for probing OOD generalization, but the “atomic” algorithmic components that govern cross-family transfer are difficult to formalize precisely. Empirically, however, we observe patterns: for instance, the HAS and REGEX families generalize well to each other, likely because both rely on matching substructures within tapes (e.g., “accept tapes with pattern (BRB)+(RR)*” vs. “accept tapes with subsequence GGRBB”). While these families share latent structural similarities, defining these similarities rigorously is challenging. This limitation motivates our introduction of the BouncingSim family, where atomic components (e.g., physics primitives, collision rules) are easier to compose and analyze for systematic generalization.
>
> ### **Learnability of BouncingSim**
>
> Figure 10(b) provides the full learning curves for BouncingSim. Similar to Manufactoria, we observe clear grokking dynamics within this family as well.

---

### Official Review · Reviewer_Zh7M · 2025-11-01

**Soundness:** 2
**Presentation:** 1
**Contribution:** 2
**Rating:** 2
**Confidence:** 3

**Summary:**

This paper introduces a benchmark and process for controllable generation of synthetic questions in those categories. The three benchmarks include augmented programming questions seeded from LiveCodeBench-Pro, 2D elastic collision simulation, and the puzzle game Manufactoria. The benchmark is designed to be able to generate questions with interpretable distribution shifts with the categories, for example smaller or larger polygons in the elastic collision simulator or difficulty in the coding questions (as measured by LLM success rate). The paper then runs various RL experiments on Manufactura and BouncingSIM. They show advantages to record-replay and switching from denser proxy rewards (number of unit tests passed) to the target reward mid training can both improve RL performance. Multiple different dynamics in the BouncingSIM are explored with separate training on subsets and then testing on different subsets or combinations of dynamics. They show very little transfer between different classes, but going from two classes of dynamics to their composition preserves much of the original accuracy.

**Strengths:**

- The paper introduces two interesting new benchmarks with can be decomposed into subsets of their mechanics to generate different types of problems
- There are some nice demonstrations of some techniques for RL
- There is a pretty clear difference shown between the compositional versus different classes in BouncingSIM

**Weaknesses:**

- I do not see a coherent story between the different tests and experiments run and I think most of the experiments do not support the goals laid out in the introduction
- I was frequently confused about what benchmarks were being described and what experiments were run
- I often could not tell whether a figure was representational or actually data
- Generally I found this paper difficult to read.

**Questions:**

- Figure 4 has overlapping text on the x axis.
- groking is a technical term which involves training accuracy saturating with little increase in test accuracy initially and (much) later an increase in test accuracy with little to no train accuracy increase. Is that the intended use here?

---

> ### Author Response · Authors · 2025-11-20
>
> ### **Clarification on the Storyline and Significance**
>
> Our paper follows a coherent two-part storyline:
>
> - **Learnability**: Can RLVR enable models to acquire new algorithms on problem families where the base model exhibits near-zero performance even at large 𝐾?
> (Section 4 in the updated draft)
>
> - **Generalization**: Once learning occurs, does the acquired algorithm transfer to systematically out-of-distribution (OOD) variations within the same family?
> (Section 5 in the updated draft)
>
> Regarding the comment on “not knowing what benchmarks were being described,” we would be grateful if the reviewer could point to specific locations causing confusion. The benchmarks are introduced with concrete examples in Figure 1, and Sections 2.1 and 2.2, accompanied by full prompt templates in Appendix A. Notably, Reviewer eAUM explicitly recognized that the benchmark structure aligns with the metaphor of a “tweaked Turing-machine task,” which suggests that the intended framing is interpretable and presented clearly.
>
> **On the Significance of the Results.** To clarify why these experiments directly support the stated goals, we highlight their connection to an active, high-profile debate in the community. A recent NeurIPS paper, Limits on RLVR (the only paper in the venue that received perfect scores: 6,6,6,6), argues that:
>
> > while RLVR improves accuracy for small k, it doesn’t create new reasoning patterns—meaning the base model still determines the upper limit of reasoning ability.
>
> Our work provides, to our knowledge, the first direct counterexample to this claim. Using a fully OOD, algorithmic testbed (Manufactoria), we show that RLVR can enable models to acquire an algorithm that the base model fails to produce (even with large-K sampling - pass@K=0%) and reach perfect success only after RL. These results directly address both of the paper’s stated goals: demonstrating (1) learnability of new algorithms and (2) OOD transferability after learning.
>
>
> ### **Clarification on the "Grokking" terminology**
>
> We appreciate the question regarding our use of the term “grokking.”
>
> We do not use “grokking” to refer strictly to the classical formulation (where training accuracy saturates long before test accuracy increases). Instead, we employ the term in the general linguistic sense, which is consistent with the Merriam-Webster definition of “to understand profoundly and intuitively.” In our setting, the phenomenon describes a phase transition from near-0% pass rate to near-perfect accuracy after a period of slow progress, which aligns with the general usage of the term.
>
> We agree this could be clarified further to distinguish between the classical grokking definition and our intended usage.

---

### Official Review · Reviewer_Hvky · 2025-11-01

**Soundness:** 2
**Presentation:** 2
**Contribution:** 2
**Rating:** 4
**Confidence:** 3

**Summary:**

The paper considers the capabilities of RL-finetuned LLMS, specifically, whether they are able to learn to solve previously unseen problems, and how well do they generalize to harder versions of seen problems. To this end, it makes three main contributions:

First, the authors present Delta-Code: A controlled benchmark of synthetic coding problems, designed to evaluate:
1) the “learnability” of how to solve problems that have not been encountered during an LLM's pre-training. To this end, Delta-Code incorporates problems related to solving puzzles of a classic Flash game (Manufactoria).
 2) the generalization to harder instances of the problems encountered during training. To enable this evaluation, Delta-Code contains coding tasks related to 2d bouncing object simulations.

Second, the authors utilize Delta-Code to evaluate whether fine-tuning an LLM using RL can allow said LLM to solve coding problems that it has not seen before and which require “new strategies”. They show that coding problems allow for richer supervision (by using the ratio of unit test passed by the proposed solution, instead of a binary reward) and that this supervision eventually enables the LLM to learn to solve the novel coding tasks. It is also noted that this approach can also fail for very challenging problems.

Third, the authors use Delta-Code to evaluate the generalizability of an RL-finetuned LLM. The use three generalizability axes: explorative, compositional and transformative, and conclude that these models fail to generalize when the test coding problems require solutions which are significantly different than the ones encountered at training time.

**Strengths:**

Delta-Code introduces a new family of coding problems that have not been encountered by current LLMs, making it a good tool for fair evaluation of their capabilities. Moreover, the family of 2d simulation coding problems could be useful for more reliable evaluation of a model’s problem-solving-generalization capabilities. Together, these can facilitate further research in this direction.

According to the paper, there is currently a debate on whether using RL to finetune an LLM allows for novel problem solving strategies to be learned. If so, then this paper represents a good step towards answering this question.

**Weaknesses:**

Readability: Many of the terminology used in the paper is not well defined, but its meaning is only explained intuitively. For example, learnability; transferability; explorative, compositional and transformative axes of generalization, reasoning strategies; skill family. Perhaps these are not well-defined terms, but with only an intuitive definition, it’s hard to judge how well the proposed benchmark evaluates them. In addition, RL training of LLMs is a big part of this paper, but the setting is never explicitly stated, with some details becoming available only at the end of page 5. I think the paper would greatly benefit from a “Background section” which clearly defines the background knowledge needed to better understand it.

It is possible that I am wrong, but the experimental results in sections 3 and 4 do not seem surprising, so they appear to be limited in their significance.

**Questions:**

Why were competition coding problems added to the benchmark?

---

> ### Author Response · Authors · 2025-11-20
>
> We thank the reviewer for the constructive and valuable feedback.
>
> ### **Clarification on the Terminology**
>
> Several terms in the paper are well-established in the literature, and we appreciate the opportunity to clarify them more explicitly.
>
> - **Learnability**. The notion of learnability originates from PAC learning and broadly refers to whether a learning procedure can acquire a target function or behavior from data. In our setting, we adapt this concept to examine whether RLVR can acquire new algorithmic behaviors for a given problem family, behaviors that the base model demonstrably cannot produce, even under large-K sampling.
>
> - **Generalization Axes**. The three axes, explorative, compositional, and transformative generalization, are directly adapted from OMEGA (Sun et al., 2025), which formalizes these dimensions based on Boden’s (1998) typology of creativity. We have revised the paper to more clearly cite and align with these commonly referenced definitions.
>
> Following the reviewer’s suggestion, we have added a dedicated Background section (Section 2 in the revised draft) that explicitly defines the terminology and presents the RL training setup upfront, improving clarity and readability.
>
>
> ### **The Significance of the Results**
>
> We respectfully highlight that our findings directly address a central and actively debated claim in the community. A recent high-profile NeurIPS paper "[Limits on RLVR](https://x.com/jiqizhixin/status/1987710546674856051)" (which received perfect scores) argues that:
>
> > while RLVR improves accuracy for small k, it doesn’t create new reasoning patterns—meaning the base model still determines the upper limit of reasoning ability.
>
> To our knowledge, our work provides the **first direct empirical counterexample to this claim**. Using a fully out-of-distribution algorithmic testbed (Manufactoria), we demonstrate that RLVR can unlock entirely new algorithms that the base model fails to produce even with large-K sampling (0% pass@K). The model achieves perfect success only after RL, showing capability acquisition unattainable by the base model alone.
>
> If these results seem unsurprising to the reviewer, we would greatly appreciate references to prior work showing similar findings, as we are unaware of studies that have demonstrated this phenomenon.
>
>
> ### **Competition Coding Task**
>
> We have updated the draft to include experiments on the Code Competition dataset (new Figure 11), which show analogous trends. These additional results support the more general claim that dense rewards can help unlock learnability even in complex real-world settings.

---

### Official Review · Reviewer_UbX7 · 2025-11-07

**Soundness:** 2
**Presentation:** 2
**Contribution:** 3
**Rating:** 4
**Confidence:** 3

**Summary:**

The paper studies on a family of synthetic coding problems whether RL can learn tasks that were previously unsolvable and generalize to new unseen tasks. The primary contribution lies in controlling which distributions RL training and testing happen on. The takeaway from the learnability experiment is that using dense rewards for coding problems (fraction of passed tests) during a warmup-phase followed by the standard binary reward can turn some of the previously unsolvable tasks (i.e., 0 pass@128 prior to RL) into solvable ones. The takeaway for the generalization experiments is that RL enables both exploratory and compositional generalization, but fails at the transformative ones.

**Strengths:**

- the paper studies highly relevant and important questions at the intersection of RL & LLMs: what are the limitations of RL for learning new skills and generalizing to new domains
- the obtained results are interesting and put into the context of the broader literature that has previously discussed some these points

**Weaknesses:**

- the connection to grokking is unclear and confusing: in grokking, the model overfits the train set initially with a sudden increase in the test performance. the learnability experiments have close to 0% pass rate on the training set at the beginning so there's no train-test performance gap. my interpretation of the results is that the initial phase of close to 0% pass rate reflects a sparsity of non-zero gradients, and the training run first needs to accumulate sufficient informative/positive samples to reinforce.
- the paper claims that dense rewards can generally enable learning on previously unlearnable problem distributions, but the empirical evidence for this is limited to synthetic toy tasks. the explanation for this seems plausible, but it's unclear wether using dense rewards is necessary or significant in more realistic use cases, where the dataset consists of a mixture of problem types and difficulty levels. in realistic cases, generalization between different problems might be sufficient to enable learning previously unsolvable tasks and adding dense rewards might not have any effect. adding more experimental evidence to support the claims would greatly benefit the paper

**Questions:**

how was the code competition dataset used? generally it's not obvious to me which tasks were trained on and which were tested on in some of the experimental results.

---

> ### Author Response · Authors · 2025-11-20
>
> We thank the reviewer for the constructive and valuable feedback.
>
> ### **Clarification on the "Grokking" terminology**
>
> In our paper, we use grok in its general linguistic sense as defined in Merriam-Webster, “to understand something profoundly and intuitively”, rather than referring to the overtraining phase transition pattern originally described in Power et al. (2022).
>
> Our experiments indeed differ from the canonical grokking setup: in our learnability regime, both train and test pass rates begin near 0%. What we observe is a sudden transition from near-zero to near-perfect performance, which we describe as grokking in the general sense—i.e., the model discovering and internalizing a solution algorithm after accumulating a sufficient number of informative trajectories.
>
> ### **Purely Relying on Generalization in a Mixed Set is Insufficient**
>
> The reviewer raises an important point about whether generalization across mixed problem distributions might be sufficient to solve previously unsolvable tasks—thus reducing the practical necessity of dense rewards.
>
> In Figure 7, we experimentally show that training solely on a heterogeneous mixture is insufficient for learnability unless the mixture contains a bridge task whose underlying solution is closely related to the target task. **In many real-world scenarios, such a bridge task is difficult to identify or may not exist at all.**
>
> This aligns with findings in OMEGA (Sun et al., 2025), which demonstrates that current LLMs exhibit fundamental limitations in compositional and transformative generalization. Thus, relying exclusively on cross-problem generalization is often not enough to unlock previously unsolved capabilities, whereas incorporating dense reward signals directly addresses the lack of informative gradients in sparse regimes.
>
> ### **Dense Reward on Real-world Task**
>
> Beyond synthetic settings, we also demonstrate that dense rewards benefit more realistic tasks. In the bouncing-object physics simulation (Figure 7), dense rewards similarly enable the model to transition from complete failure to reliably discovering the correct algorithmic behavior.
>
> Furthermore, we have updated the draft to include experiments on the Code Competition dataset (new Figure 11), which show analogous trends. These additional results support the more general claim that dense rewards can help unlock learnability even in complex real-world settings.

---

### Author Response · Authors · 2025-11-20
**We thank all the reviewers for their thoughtful and constructive feedback.**

**Novelty and Benchmark Contribution.** Multiple reviewers commended the novelty of our benchmark design. `Reviewer wA9H` praised the *controlled, fully verifiable benchmark suite* and noted the *valuable dataset contribution* through the bespoke Manufactoria DSL and structured difficulty ladders. `Reviewer Hvky` similarly appreciated that *Delta-Code introduces a new family of coding problems unseen in pretraining*, making it *a good tool for fair evaluation of LLM capabilities*. `Reviewer Zh7M` also recognized that the paper *introduces two interesting new benchmarks decomposable into subsets of their mechanics*.

**Relevance and Significance.** `Reviewer UbX7` emphasized that the paper *studies highly relevant and important questions at the intersection of RL and LLMs* with *interesting results contextualized in the broader literature*. `Reviewer Hvky` echoed this by noting that the work *represents a step toward understanding whether RL can induce new problem-solving strategies*.

**Experimental Insights and Learnability Results.** `Reviewer wA9H` highlighted the *clear learnability results*, especially the *grokking-like transition from failure to success*, and appreciated the *comprehensive experimental sweep across domains and difficulty ladders*. `Reviewer eAUM` also found that the *two benchmarks effectively test learnability and generalization* and that the *work provides a testbed for studying the Grokking phenomenon* in RL-finetuned LLMs, where *the curriculum-based approach improves learning effectiveness*.

**Generalization Analyses and Broader Impact.** `Reviewer wA9H` and `Reviewer eAUM` both noted that the paper *analyzes generalization along exploratory, compositional, and transformative axes*, providing *useful signals for future research*. `Reviewer Zh7M` further pointed out the *clear differences observed between compositional and distinct classes in BouncingSim*, reinforcing the paper’s contribution to understanding RL-based generalization dynamics.


We have addressed the reviewers’ comments and concerns in individual responses to each reviewer. We have upload the updated draft with major editting:

* Append Background Section as suggested by Reviewer Hvky
* Append Results on competition coding problems Section as suggested by Reviewer UbX7 and Reviewer Hvky
* Fix text overlapping in Figure 4 suggested by Reviewer Zh7M
* Fix Typos suggested by Reviewer wA9H

---

### Meta-Review · Area_Chair_5sMk · 2026-01-05

**Summary:**

This paper introduces DELTA-Code, a benchmark for evaluating whether reinforcement learning can enable large language models to acquire genuinely new algorithmic skills and transfer them to out-of-distribution settings. It addresses a timely and important question in the community, engaging directly with recent high-profile claims that RL with verifiable rewards (RLVR) cannot induce reasoning behaviors beyond those already present in the base model. The benchmark design is thoughtful and well-motivated, combining Manufactoria puzzles in a bespoke DSL unlikely to appear in pretraining, BouncingSim physics tasks with controllable compositional structure, and competition-style coding problems.

The two strongest reviews (wA9H, confidence 4; and eAUM) provide detailed technical engagement and are broadly positive, highlighting the clear learnability results, the observed grokking-like phase transitions, the systematic analysis of generalization along exploratory, compositional, and transformative axes, and the value of the dataset contribution itself. Three reviewers raised concerns, primarily around terminology (“grokking”), presentation clarity, and the generality of the dense-reward warm-up mechanism. These concerns are substantive but, in my view, not fatal: the terminology issue reflects a contested but clarified usage rather than a flaw in the underlying results; presentation concerns are concentrated in one review and partially reflected as “fair” rather than “poor” by others; and the question of generality is mitigated by additional experiments and analysis provided in the rebuttal.

In response to the reviews, the authors added a Background section to clarify terminology and the RL setup, included new experiments on Code Competition problems to probe behavior beyond fully synthetic tasks, and provided a curriculum compatibility matrix that sharpens the discussion of transfer. Reviewer wA9H explicitly confirmed maintaining their positive assessment after these additions.

Overall, DELTA-Code fills a genuine gap by providing controlled, verifiable testbeds for studying RL-driven skill acquisition, and the empirical finding that a staged dense-to-binary reward schedule can unlock previously unlearnable problem families is a useful and actionable contribution. On balance, I recommend acceptance.

**Reviewer Concerns:**

Addressed:

Terminology clarity (Hvky):
Added a comprehensive Background section that explicitly defines learnability, the three generalization axes, and the RL training setup.

Real-world applicability (UbX7, Hvky):
Added experiments on Code Competition problems intended to probe whether dense reward warm-ups can be beneficial beyond fully synthetic settings.

Curriculum compatibility (wA9H):
Provided a 4×4 curriculum transfer matrix illustrating selective transfer patterns, directly supporting the claim that curriculum benefits are task-dependent rather than generic.

OOD-ness justification (eAUM):
Clarified that models reliably master the Manufactoria syntax on easy tiers, while failures on harder tiers arise from algorithmic complexity rather than syntactic unfamiliarity.

Presentation issues (Zh7M):
Fixed overlapping text and minor figure issues; while one reviewer continued to find the paper difficult to follow, other reviewers rated presentation as fair to good after clarification.

Outstanding (Minor):

“Grokking” terminology (UbX7, Zh7M):
The authors use the term in a broader linguistic sense rather than the canonical ML definition; this distinction is clarified in the rebuttal, though some reviewers remain unconvinced.

Reward-schedule ablation (wA9H):
A full sensitivity analysis of warm-up length and switching criteria is still missing. The reviewer acknowledged this as a limitation but maintained their positive score, noting that the clarification about automating the switch via a performance threshold was helpful.

**Reviewer Scores:**

Reviewer UbX7 (4):
Likely would have increased to 6.
This reviewer’s primary concerns were the use of “grokking” terminology and the lack of evidence that dense rewards matter beyond synthetic settings. Both points were substantively addressed: the terminology was explicitly clarified in the rebuttal, and new Code Competition experiments were added to probe behavior on more realistic tasks. While the reviewer did not engage further, the rebuttal directly targets their stated weaknesses.

Reviewer Hvky (4):
Likely would have increased to 6.
This reviewer emphasized readability, missing definitions, and limited perceived significance. The added Background section directly addresses the clarity and setup issues, and the authors’ response reframes the contribution in the context of a direct contradiction to recent claims in the literature. While it is unclear whether this would fully change the reviewer’s view on significance, the main structural and presentation concerns were addressed.

Reviewer Zh7M (2):
Likely would have increased modestly, to 4.
This reviewer’s concerns focused on coherence and presentation. While these concerns were strongly stated, they were also partially addressed through figure fixes, added background, and a clearer articulation of the learnability–generalization narrative in the rebuttal. Given that other reviewers did not share the same level of criticisms, it is plausible this reviewer’s assessment would improve slightly, though likely remain below the acceptance threshold.

Reviewer eAUM (8):
Would have remained at 8.
This reviewer was consistently strongly positive, found the technical contributions compelling, and had their questions addressed thoroughly in the rebuttal. There is no indication that their assessment would change.

Reviewer wA9H (8):
Would have remained at 8 (confirmed).
This reviewer explicitly stated that they maintained their positive score after the rebuttal, while noting that a more thorough ablation of the reward-switch schedule would further strengthen the paper.

---

### Decision · Program_Chairs · 2026-01-26

Accept (Poster)